# Advanced biofilm analysis in streams receiving organic deicer runoff

**Michelle A. Nott**[1]\*, **Heather E. Driscoll**[2], **Minoru Takeda**[3], **Mahesh Vangala**[4¤], **Steven R. Corsi**[1], **Scott W. Tighe**[5]

**1** Upper Midwest Water Science Center, U.S. Geological Survey, Middleton, Wisconsin, United States of America, **2** Vermont Genetics Network, Department of Biology, Norwich University, Northfield, Vermont, United States of America, **3** Graduate School of Engineering, Yokohama National University, Hodogaya, Yokohama, Japan, **4** Vermont Genetics Network, University of Vermont, Burlington, Vermont, United States of America, **5** Advanced Genome Technologies Core, University of Vermont, Burlington, Vermont, United States of America

¤ Current address: Data Sciences and Technology, University of Massachusetts Medical School, Worcester, Massachusetts, United States of America

\* mnott@usgs.gov

**Data Availability Statement:** All non-sequence data are available in a companion USGS Data Release report, located here: https://doi.org/10.5066/F75H7DFS. Sequence (i.e., MPS and isolate) and microarray data are housed on the NCBI

## Abstract

Prolific heterotrophic biofilm growth is a common occurrence in airport receiving streams containing deicers and anti-icers, which are composed of low-molecular weight organic compounds. This study investigated biofilm spatiotemporal patterns and responses to concurrent and antecedent (i.e., preceding biofilm sampling) environmental conditions at stream sites upstream and downstream from Milwaukee Mitchell International Airport in Milwaukee, Wisconsin, during two deicing seasons (2009–2010; 2010–2011). Biofilm abundance and community composition were investigated along spatial and temporal gradients using field surveys and microarray analyses, respectively. Given the recognized role of *Sphaerotilus* in organically enriched environments, additional analyses were pursued to specifically characterize its abundance: a consensus *sthA* sequence was determined via comparison of whole metagenome sequences with a previously identified *sthA* sequence, the primers developed for this gene were used to characterize relative *Sphaerotilus* abundance using quantitative real-time PCR, and a *Sphaerotilus* strain was isolated to validate the determined *sthA* sequence. Results indicated that biofilm abundance was stimulated by elevated antecedent chemical oxygen demand concentrations, a surrogate for deicer concentrations, with minimal biofilm volumes observed when antecedent chemical oxygen demand concentrations remained below 48 mg/L. Biofilms were composed of diverse communities (including sheathed bacterium *Thiothrix*) whose composition appeared to shift in relation to antecedent temperature and chemical oxygen demand. The relative abundance of *sthA* correlated most strongly with heterotrophic biofilm volume (positive) and dissolved oxygen (negative), indicating that *Sphaerotilus* was likely a consistent biofilm member and thrived under low oxygen conditions. Additional investigations identified the isolate as a new strain of *Sphaerotilus montanus* (strain KMKE) able to use deicer components as carbon sources and found that stream dissolved oxygen concentrations related inversely to biofilm volume as well as to antecedent temperature and chemical oxygen demand. The airport

platform (in SRA, Nucleotide, and GEO, respectively) under BioProject PRJNA360543 (https://www.ncbi.nlm.nih.gov/bioproject/?term= PRJNA360543). All records are public now. Direct accession numbers are as follows for SRA: SRX2476746 and SRX2476747; for Nucleotide: KP096714.1, KP096715.1, and KF614510.1; for GEO (Series GSE129990): GSM3729025, GSM3729026, GSM3729027, GSM3729028, GSM3729029, GSM3729030, GSM3729031, GSM3729032, GSM3729033, GSM3729034, and GSM3729035.

**Funding:** Financial support for this research was provided by Milwaukee Mitchell International Airport (https://www.mitchellairport.com/) and the US Geological Survey (https://www.usgs.gov/) and was received by SRC. Funding for Vermont Genetics Network (VGN) Bioinformatics and Microarray Core services performed by HED was supported by an Institutional Development Award (IDeA) from the National Institute of General Medical Sciences (NIGMS; https://www.nigms.nih. gov/) of the National Institutes of Health (NIH) under grant number P20GM103449; the grant was received by Rex Forehand at The University of Vermont. The funders had no role in study design, data collection and analysis, decision to publish, or preparation of the manuscript.

**Competing interests:** The authors have declared that no competing interests exist.

setting provides insight into potential consequences of widescale adoption of organic deicers for roadway deicing.

## Introduction

Organic deicers are increasingly making their way into more widespread roadway application due to observed performance enhancements (over salt alone) and increased awareness of the persistent ecological effects of road salt application [1–5]. Within the roadway deicing community, the effects of organic deicers on short-term dissolved oxygen (DO) in nearby waterways are generally recognized; however, the potential for biofilm proliferation is not. The consistent, long-term use of organic deicers within airport settings makes their receiving streams useful ecosystems for studying a broad range of potential effects of widespread organic deicer use on aquatic systems.

Airfield pavement and aircraft deicers are frequently applied to aircraft surfaces and airfield pavements to enhance the safety of air travel in cold weather. The freezing point depressants in airfield pavement deicing materials (PDMs) and aircraft deicer and anti-icer fluids (ADAFs) are low-molecular-weight organic compounds that are miscible in water and can be readily consumed by heterotrophic bacteria (i.e., bacteria requiring external organic compounds for growth) in the environment [6]. As a result, the oxygen demands posed by deicers are high. At the time of this study (i.e., 2009–2014), the deicer and anti-icer products used at Milwaukee Mitchell International Airport (MMIA) ranged in chemical oxygen demand (COD) from 250,000 to 818,000 mg/L as applied formulations [7], and included liquid potassium acetate PDM, propylene glycol (PG) Type I aircraft deicer, and PG Type IV aircraft anti-icer. To provide context, the typical COD of untreated domestic sewage is many orders of magnitude lower, at approximately 750 mg/L [8]. Airports commonly administer runoff management programs to recover deicers before they reach receiving waters. Effectiveness of these programs varies, but can be as high as 60% in overall ADAF collection efficiency [9]. Much of the uncollected ADAFs (and PDMs) enter into surface water systems, while some degrade on airfield surfaces or move into groundwater systems [10]. The strong oxygen demands exerted by decay of these chemicals strain the absorptive capacity of the small receiving streams that typically drain airports [10]. This oxygen depletion, coupled with toxicological effects observed from PDMs and ADAFs [10], create a challenging environment for maintenance of healthy aquatic communities. For many receiving streams, this challenge is further compounded, and in some cases potentially superseded, by the degradation of habitat, water quality, and biological communities associated with drainage of an urban watershed [11–13].

In waters receiving external organic carbon inputs, the presence of biofilms dominated by bacteria in the *Sphaerotilus-Leptothrix* group is common and has been noted for nearly two centuries [14–19]. Literature in the more recent past has noted the presence of these organisms and their associated biofilms in streams receiving effluent from airports that conduct deicing, paper mills, dairy factories, and slaughterhouses, as well as within wastewater treatment plants (as an agent in activated sludge bulking) [20–22]. The *Sphaerotilus-Leptothrix* group is a small group of organisms that are both genetically and phenotypically similar [23]. However, only two of its members have been shown to exhibit growth stimulation in nutrient-rich environments: *Sphaerotilus natans* and *Leptothrix cholodnii* [23]. Of these, *Sphaerotilus natans* is considered the typical organism in waters with excess organic material [24].

In a broad context, stream biofilms are ubiquitous and serve important ecological functions [25]; however, in organic-rich settings, biofilms often coat the stream bottom in thick blankets

of growth that degrade normal macroinvertebrate communities [26,27] and impede fish spawning [28]. Many airport receiving streams have been classified as impaired under Section 303(d) of the Clean Water Act [10]. The literature describes several common environmental effects on airport receiving streams related to deicing/anti-icing activities: DO depletion, fish kills, contamination of human drinking water supplies, aquatic community degradation, and aesthetic effects such as foaming, odor, and discoloration [10]. The presence of heterotrophic biofilms in these streams is intimately intertwined in many of these environmental effects, although these biofilms do not appear to have been rigorously studied in this setting.

The overall aim of this study was to enhance understanding of the environmental drivers prompting heterotrophic biofilm proliferation in a manner that could help inform prevention strategies and allow for the potential reestablishment of healthier and more diverse aquatic communities in streams receiving organic deicer runoff. The first study objective was to systematically characterize spatial and temporal patterns in biofilm abundance and community composition. The second study objective was to characterize biofilm response to environmental conditions in terms of (1) biofilm abundance, (2) biofilm community composition, and (3) the relative abundance of *Sphaerotilus* within the biofilm community. Expectations were that (1) biofilm abundances would be enhanced when COD concentrations were high, (2) that community composition would vary along COD and temperature gradients, and that (3) representation of *Sphaerotilus* in the biofilm would be highest when COD concentrations were high. This study utilized a field- and laboratory-based approach. Given the likely influence of deicer contributions, sites were selected upstream and downstream from the airport to maximize differences in organic loading to the different sites. Samples were collected along spatial and temporal gradients: biofilm field surveys were used to characterize abundance; microarray was used to characterize biofilm community composition; and quantitative real-time polymerase chain reaction (qPCR) was used to investigate the relative abundance of *Sphaerotilus* within biofilms. This additional characterization of *Sphaerotilus* abundance was performed due to a combined expectation of their importance within the biofilm (based on literature) and the dearth of sequence representation in public databases (and on the microarray chip). Water-quality data, including grab and flow-weighted composite samples, were collected throughout both deicing seasons, and regressions and correlations were used to explore relations with biofilm metrics.

## Materials and methods

### Sampling locations and frequency

In an effort to assess differences along a spatial gradient, four primary sites were selected on two streams surrounding the airport grounds. The most upstream site (US1) was located on Edgerton Channel, upstream from the airport; the remaining three sites (DS1, DS2, and DS3) were located downstream from the airport on Wilson Park Creek (S1 Table; Fig 1). Edgerton Channel flows into Wilson Park Creek approximately 23 meters downstream from the DS1 site. Both streams drain urban settings and are small watersheds, with drainage areas to sites ranging from 2.1 to 30.6 square kilometers. Data collection also occurred at two stream gages near the DS1 and DS3 sites on Wilson Park Creek (S1 Table; Fig 1).

Sites DS1 and DS1-gage were both located within MMIA grounds; approval for access to the sites and sample collection from the sites was requested and granted by MMIA. All other sites were in unrestricted stream reaches and were accessed using public bridge crossings or with the permission of local land owners. No protected species were sampled.

Sampling trips were conducted during low-flow periods at each of the four main sites. In an effort to assess differences along a temporal gradient, sampling occurred approximately

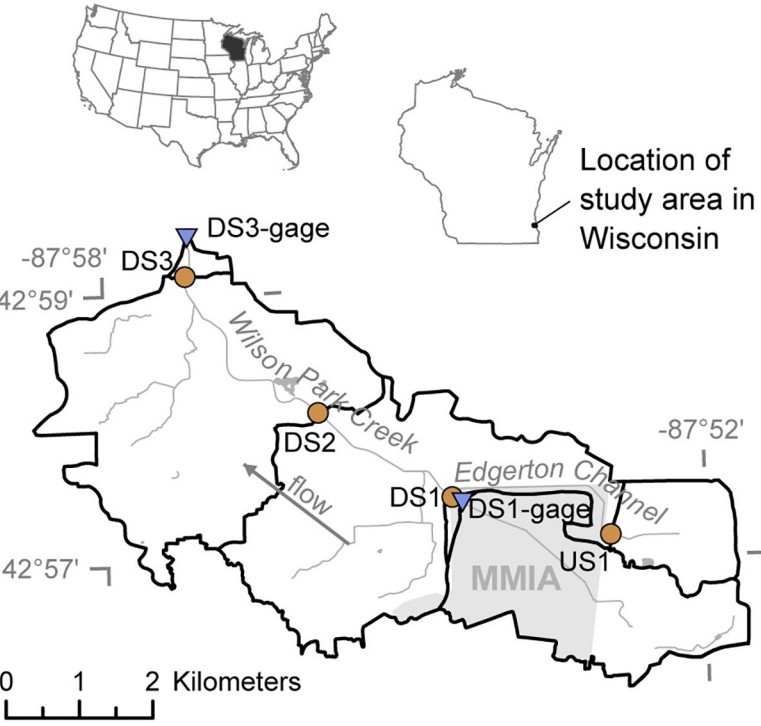

**Fig 1. Location of study area and sampling sites.** Brown circles indicate biofilm sampling sites and blue triangles indicate stream gages. Light gray area represents Milwaukee Mitchell International Airport (MMIA) grounds.

monthly throughout two deicing seasons (December 2009–June 2010 and November 2010–June 2011), with twice-monthly sampling during 2 months in the spring/early summer. Additional, continuous water-quality measurements and samples were collected during this time at the two nearby stream gages. Given study objectives to characterize the effects of deicers and anti-icers, sampling was primarily done during months with deicer influence, with a smaller subset of samples collected during months without deicer influence for the purpose of comparison. Deicing activity typically starts in late November and continues into April; stream COD concentrations often remain elevated into May due to deicer contributions from the shallow groundwater system. Samples collected in November of 2010 preceded deicer application, so these samples were considered not to have deicer influence. Samples collected from December through May were considered deicer influenced.

## Water data

Water quality data were collected to document environmental conditions in the stream, and to investigate the environmental factors potentially affecting biofilms. Water-quality samples were collected during each trip as grab samples from the centroid of flow. Samples were analyzed for nutrients (dissolved nitrate+nitrite, total Kjeldahl nitrogen (TKN), and total phosphorus) and COD. All water-quality analyses were performed by the Wisconsin State Laboratory of Hygiene (WSLH) in accordance with standard analytical methods: USEPA 353.2 for nitrate+nitrite, USEPA 351.2 for TKN, USEPA 365.1 for total phosphorus, and ASTM D1252-95(B) for COD [29,30]. Field properties were measured during each trip using a portable, multiparameter sonde (YSI Incorporated, Yellow Springs, Ohio, USA) and included temperature, specific conductance (SC), DO, and pH. Field properties were typically measured

at each site during the same general time of day to minimize the effect of diurnal fluctuations across sampling trips.

Year-round, continuous monitoring data for streamflow and temperature were collected at streamgages near the DS1 and DS3 sites. Streamflow was determined using standard methods [31]. Calibration of the continuous temperature monitors were checked against a pre-calibrated multiparameter sonde. At the remaining two sites, US1 and DS2, temperature sensors were used to measure temperature every 15 minutes throughout both deicing seasons (Onset Computer Corporation, Bourne, Massachusetts, USA). Temperature data from all sites were corrected to remove erroneous readings and were stored in the USGS Automated Data Processing System (ADAPS) database. At the DS1-gage site, weekly flow-weighted composite COD and intermittent (n = 11) deicer samples (i.e., acetate and PG, the two main freezing point depressants utilized at MMIA) were collected throughout the two deicing seasons; analyses were performed by the WSLH. Acetate was analyzed on the DIONEX AS15 separator column (DIONEX, Sunnyvale, California, USA) per manufacturer instructions [12]. PG was analyzed using USEPA 8015C [32].

Temperature data for all four sites were retrieved from the USGS ADAPS database as 15-minute data. At each of the ungaged sites, data from the two thermocouples were averaged. Averaged data showed strong relations with data collected at nearby gaged sites (US1 and DS1-gage: $Temperature_{US1} = 1.13 * Temperature_{DS1-gage} - 1.71$, $R^2 = 0.85$; and DS2 and DS3-gage: $Temperature_{DS2} = 1.01 * Temperature_{DS3-gage} + 0.16$, $R^2 = 0.98$), and determined relations were used to fill in gaps in the records at the ungaged sites. Various temperature statistics (mean, maximum, minimum, median, and standard deviation) were calculated for multiple time windows (0.5, 1, 2, 4, 6, 8, 12, 16, and 20 weeks) preceding each biofilm sample.

Likewise, detailed COD data were only available at the DS1-gage site, and data from this site were used to estimate concentrations at downstream sites through a series of steps. First, the time required for water to flow from DS1-gage to DS3-gage was calculated for each flow composite sample, using the following equation:

$$T = 29.35 * Q_{DS3-gage}^{-0.62} \tag{1}$$

Where $T$ is the time of travel between DS1-gage and DS3-gage, in hours, and $Q_{DS3-gage}$ is the streamflow value nearest the event midpoint at DS3-gage. This flow-dependent relation was determined previously for these two sites through dye-tracer studies (methods are described in the Supporting Information). Next, the volume of water associated with the time period represented by each flow-composite sample at each gage was calculated. Associated timing and volume estimates at DS2 were calculated through drainage area scaling of data from DS1-gage and DS3-gage using the following equation:

$$V_{DS2} = \left( \left( \frac{V_{DS1}}{DA_{DS1}} + \frac{V_{DS3}}{DA_{DS3}} \right) \Big/ 2 \right) * DA_{DS2} \tag{2}$$

Where $V$ is the event volume and $DA$ is the drainage area at the sites indicated in subscripted text. The load associated with each flow-composite event at DS1-gage was calculated (as event volume * concentration); the load associated with each event was then used to calculate associated concentrations at the downstream sites (using DS1-gage load/downstream site event volume). This flow-composite dataset was then used as the basis for calculating flow-weighted mean concentrations and loads for multiple antecedent time windows (2, 4, 6, 8, 12, 16, and 20 weeks) preceding each biofilm sample. Calculations utilized the middle date/time of the flow-composite sample. For DS1-gage and DS3-gage, measured (variable time-step) streamflow data were used. For DS2, streamflow was estimated (from 15-minute time-step data) using

drainage area scaling of data from DS1-gage and DS3-gage using the following equation:

$$Q_{DS2} = \left( \left( \frac{Q_{DS1}}{DA_{DS1}} + \frac{Q_{DS3}}{DA_{DS3}} \right) \Big/ 2 \right) * DA_{DS2} \tag{3}$$

Where $Q$ is streamflow and $DA$ is drainage area at the sites indicated in subscripted text.

Quality control (QC) samples were collected in association with both point and flow-weighted composite samples, and results are described in the Supporting Information. All water-quality (regular and QC), streamflow, and temperature data are available elsewhere [33].

## Biofilm field surveys

Biofilm field surveys were done in order to systematically document observable biofilm prevalence in the streams and explore changes in biofilm volume with relation to time, space, and the effects of environmental factors. Data collection approaches were adapted from established rapid periphyton survey and stream habitat protocols [34–36]. Data were collected at multiple spatial scales: reach, transect, and transect point. Briefly, a short reach was established at each of the four primary sites and revisited each sampling trip as conditions allowed; each reach contained five equidistant transects along which data were collected describing biofilm and physical stream channel characteristics from approximately 50 (total) points. Additional details on biofilm field survey methods can be found in the Supporting Information.

Using data collected during field surveys, biofilms were categorized into one of four operational classes according to dominant color and morphology: 'soft algae', 'transition' (soft algae-heterotroph mix), 'heterotrophs', and 'diatoms' (Fig A in S1 Appendix). This study focused on heterotrophic biofilms, operationally defined for surveying purposes as biofilms without visual algal representation. To allow for comparisons between sites, biofilm volumes have been standardized to a 50 square-meter reach. Calculations of biofilm volume were performed for each class of organisms (example here is for heterotrophs) in each sample using the following equation:

$$V_{Heterotrophs} = (F_{Heterotrophs} * 50) * T_{Heterotrophs} \tag{4}$$

Where $V$ is biofilm volume, $F$ is the fraction of survey points, 50 is the (standardized) reach area, and $T$ is the median biofilm thickness measured at survey points categorized into the specified class of organisms. In samples where heterotrophic biofilms were not observed, volumes calculated to zero due to a zero value for $F$. Raw and aggregated biofilm field survey data are available elsewhere [33].

## Regressions

Linear regression analyses were performed to more fully understand the relation of organic measures collected during the study. First, regressions (in log10 space) were run between concurrently collected COD and total deicer (i.e., acetate and PG) concentrations to explore the utility of using COD as a surrogate for deicer concentrations. Second, regressions (in log10 space) were run between grab COD and the nearest flow-composite COD concentrations to determine how similar grab COD and measured (at DS1) or estimated (at DS2 and DS3) flow-composite COD concentrations were to each other; regressions were done by site and across sites.

To explore factors affecting heterotrophic biofilm volume and DO, regression analysis was performed using a suite of environmental measures as predictors (as defined in S2 Table). Notably, DO was not used in the biofilm regression because the presence of biofilms could

substantially affect DO. Linear regression models were estimated using stepwise ordinary least squares regression with forward and backward selection; variable selection within the stepwise regression was based upon minimization of the Bayesian Information Criterion. Data analyses were done using the R project for statistical computing with core functionality [37]. The data and script for running these two regressions are available elsewhere [33]. To facilitate log-transformation of biofilm volume data for these regressions, zero values were substituted with a value that was half of the minimum non-zero biofilm volume. All regressions (and correlations) run for this study utilized reporting level values in the event of left-censored data (i.e., values reported as less than a reporting level).

## Biofilm sample collection and laboratory analyses

**Sample collection.** During 2009–2011, samples were collected from DS1 every sampling event, and twice each spring/early summer from several other locations including US1, stream gage lines at DS1-gage, the sandy mid-channel of DS2, the rip-rapped stream edge of DS2, and DS3 (Fig 1, Fig B in S1 Appendix, S3 Table). Samples represented composites of the major, visibly distinct (via color and structure) benthic biofilm types noted during the field survey at each site. Samples were collected by scraping the benthic substrate using a sterile, disposable petri dish and depositing into two sterile centrifuge tubes. Each day, tubes were stored on wet ice in the field, and then shipped overnight on ice packs to the Vermont Integrative Genomics Resource (VIGR) laboratory at the University of Vermont Cancer Center. Upon receipt, tubes used for microscopic analysis were stored at 4ºC, and tubes used for genetic analyses were frozen at -80ºC. The only deviation from this shipping regimen pertained to the samples collected in June of 2010: these samples were collected on a Thursday and Friday, held over the weekend at the temperatures described above, and then shipped overnight on ice packs the following Monday. Following preliminary analyses of data from these samples, more specific identification and characterization of the primary observed filamentous organism was sought. As a result, an additional sample was collected in May 2014 from a site approximately 296 meters downstream from DS1 and was used exclusively as inoculum for culturing isolates. This sample was collected using the same basic method as previous samples; however, instead of targeting the full length of major, distinct biofilm types, collection for this sample focused on just the filamentous ends of the heterotrophic biofilms.

**Microscopy.** Microscopy was performed on all samples (S3 Table) to characterize basic community composition. Particular emphasis was given to assessing the presence or absence of filamentous sheathed bacteria having morphologies consistent with *Sphaerotilus*, *Leptothrix*, and *Thiothrix* species. Samples were examined at 200, 400, and 1000x magnification using a Zeiss Axio Scope (Jena, Thuringia, Germany) using standard bright light, differential interference contrast, and epifluorescence at 485 and 525 nm excitation with long pass emission filters to distinguish chlorophyll- or phytopigment-containing organisms from sheathed bacteria. Photomicrographs were collected for all samples. QC samples were collected from each of the sites and results are described in the Supporting Information. Microscopic assessments were reported as text descriptions of sample composition; text descriptions were used to assess the presence or absence of sheathed bacteria in samples. These data are available elsewhere [33].

**DNA extractions.** Three DNA extraction approaches were tested to determine which would provide the highest DNA yields from biofilm samples: (1) hot phenol chloroform, (2) a cetyl trimethyl ammonium bromide method from Omega Bio-tek (D3373-01; Norcross, Georgia, USA), and (3) Qiagen DNA QIAamp system (Hilden, Germany). The Omega Bio-tek method showed the highest yields and was used to extract DNA from all samples using

approximately 200 mg (wet weight) of sample; extracted DNA was quantified using a Nano-Drop spectrophotometer (ND-1000; Thermo Fisher Scientific, Waltham, Massachusetts, USA) and Bioanalyzer 2100 (Agilent; Santa Clara, California, USA). Additional details are provided in the Supporting Information.

**Microarray.** Microarray techniques were used to explore community composition on a subset of 11 samples collected across spatial and temporal scales (S3 Table), using the second-generation (G2) PhyloChip (Affymetrix; Thermo Fisher Scientific) microarray platform as previously described [38]. Despite the static nature of sequences available on the chip, microarray technology continues to offer broad assessments of microbial community composition [39–42] and allowed for genus-level taxonomic resolution here. PhyloChip G2 .CEL files were analyzed by Second Genome (South San Francisco, California, USA) using an empirical approach to define unique operational taxonomic units (eOTUs) [43–45]. The criteria for scoring the probe-level fluorescence intensity (FI) and the process by which individual probes are clustered into probesets, aka eOTUs, are described in detail elsewhere [43]. The eOTU abundances from the analysis of PhyloChip data were further analyzed using MeV (MultiExperiment Viewer) in the TM4 software [46]. Hierarchical clustering of microbial genera utilized the average linkage method and Pearson Correlation distance metric [47,48]. Additionally, a principal coordinates analysis (PCoA) was performed to assess relations between samples using Fast UniFrac [49]. Additional details are provided in the Supporting Information; data are available on the NCBI Gene Expression Omnibus (GEO) repository (series: GSE129990), and in a companion data report [33].

**Whole metagenome sequencing.** Whole genome shotgun DNA sequence data were used to characterize the metagenome of two biofilm samples collected from DS1 (February 24, 2010, and March 18, 2010; S3 Table). Sequence data were obtained primarily to determine a consensus *sthA* gene sequence for primer development and subsequent exploration via qPCR; however, sequences also provided insights into community composition. Total DNA from the two samples was extracted and prepared into whole-genome sequencing libraries using the Illumina (San Diego, California, USA) TrueSeq DNA library kit. Libraries were checked for quality using a Qubit spectrofluorometer (Thermo Fisher Scientific; Waltham, Massachusetts, USA), NanoDrop spectrophotometer (Thermo Fisher), Agilent 2100 Bioanalyzer (Santa Clara, California, USA) and KAPA NGS library quantification kit (Roche; Basel, Switzerland). DNA was sequenced using a paired-end flow cell (2x100 bp) using the Illumina HiSeq 1500.

A consensus *sthA* sequence was assembled by aligning whole metagenome sequencing (WMS) data to existing *sthA* information from the NCBI sequence repository (namely, *S. natans sthA* sequence, GenBank AB050640.1) using DNASTAR SeqMan NGen 12.3.1 (Madison, Wisconsin, USA). This consensus sequence was used to develop primers using NCBI Primer-BLAST software (Bethesda, Maryland, USA) and DNASTAR SeqBuilder Pro. Due to the high GC content of the target gene, primer Tms were between 61 and 64 degrees Celsius. Primers were validated using qPCR and standard PCR assays, and the resulting 905 and 302 base pair amplicon products were validated using Sanger sequencing with the same forward and reverse primers. Developed primers were used for subsequent sample exploration using qPCR, as well as amplification and sequencing of the *sthA* gene from the cultured isolate. The final consensus sequence was compared back against the *S. natans* GenBank sequence (GenBank AB050640.1).

For community composition analyses, sequence data was converted to FASTQ format using CASAVA software (Illumina) and submitted to the One Codex (San Francisco, California, USA) software platform for microbiome profiling against the RefSeq Complete Genomes, One Codex genomes, and targeted loci databases (5S, 16S, 23S, gyrB, rpoB, 18S, 28S, and ITS genes) [50]. Results were filtered to genera having at least 3% classified reads.

Additional details are provided in the Supporting Information. Sequences have been deposited into Sequence Read Archive (SRA) (accession numbers: SRX2476746 and SRX2476747, respectively).

**Quantitative real-time PCR.** qPCR methods were used on all samples (S3 Table) to quantify the number of *sthA* and 16S copies within the biofilm to characterize the abundance of *Sphaerotilus* relative to the total bacterial population. qPCR was performed using an Applied Biosystems (Foster City, California, USA) 7900HT sequence detection system with SYBR green chemistry. Genomic copy number for DNA targets were determined using a SYBR green standard curve method derived from pure genomic bacterial DNA with a known genome copy number for normalization purposes. Careful examination of the qPCR dissociation curves was necessary because the custom primers were designed around a very difficult region of the GC-rich *sthA* gene and had the unavoidable side effect of amplifying off-target amplicons present in the sample. A total of four amplicons were noted in the qPCR data with differing thermal melting temperatures (Tms) and dissociation curves (80, 86, 88, 90ºC), but only the 80ºC and 90ºC amplicons were specific for the *sthA* gene as determined by Sanger sequencing of the qPCR products. Genomic copy numbers were adjusted to correct for the proportion of on-target amplicons in each sample. Additional details are provided in the Supporting Information.

Spearman rank correlations [51] were used to investigate relations between relative (to bacterial population) abundance of *Sphaerotilus* to environmental conditions.

QC samples were collected from each of the sites; results are described in the Supporting Information. qPCR data (for both regular and QC samples) are available elsewhere [33].

**Isolating and sequencing pure strains of sheathed bacteria.** An isolate was sought to verify the filamentous bacterium's identity (via ribosomal gene sequencing and sheath composition analyses) as well as to determine the degree of alignment with the *sthA* consensus sequence yielded from WMS. Culturing of the May 2014 sample was done on three media and examined under a dissecting microscope over the course of 3 weeks until a sheathed bacterium was observed. Subculturing was performed using a micromanipulator until pure (Fig C in S1 Appendix). A total of 23 colonies were recovered; the 16S rRNA gene of each was PCR-amplified and Sanger sequenced using two sets of standard 16S rRNA gene primers (519F/1390R and 27F/1492R; Table A in S1 Appendix; [52]). Sequence information from the 23 colonies indicated that they were all composed of the same bacterial strain.

Additional Sanger sequencing was performed to characterize the entire 16S-ITS and partial 23S rRNA gene of the ribosomal operon (16S primers: 27F, 519F, 536R, 1055F, 1330F, 1492R; and 23S primers: 37R, and 127R; Table A in S1 Appendix; [53]). The resulting sequences were used to assemble two contigs using ChromasPro (Technelysium Pty Ltd, Brisbane, Australia) and were deposited into the NCBI NR database (accession numbers: KP096714.1 and KP096715.1, respectively) (Fig D in S1 Appendix). BLAST comparisons of the 16S-ITS sequence against the NCBI nucleotide NR database showed highest max score and identity (2599 and 100%, respectively) with *Sphaerotilus montanus* strain HS (GenBank NR_116396.1), and indicated this organism was likely a new strain of *Sphaerotilus montanus*. The purified strain was deposited into the American Type Culture Collection (ATCC) as *Sphaerotilus montanus* strain KMKE (ATCC BAA-2725).

Sanger sequencing was also performed for the *sthA* gene using PCR primers designed from the WMS sequence data (primers 134F and 1041R; Table A in S1 Appendix), and the resulting sequence was deposited into NCBI GenBank as a putative glycosyltransferase (*sthA*) gene (accession number: KF614510.1). In an effort to verify the presence of the gene in the biofilm, the isolate *sthA* sequence was also aligned to the *sthA* consensus sequence determined from

WMS of environmental samples. Additional details are provided in the Supporting Information.

**Sheath composition analysis.** Sheath composition analysis was performed to further verify the identity of the isolated filamentous bacterium. The isolate and a reference organism (*Sphaerotilus natans* JCM 20382 = NBRC (IFO) 13543 = ATCC 15291) were statically cultured and the sheaths were prepared according to the method established for *S. natans* [54]. The sheath was hydrolyzed and the neutral and amino sugars released were derivatized to alditol acetates according to a previously described method [55], followed by gas chromatography (GC) under the conditions as reported previously [56]. Isolate sheath composition data are available elsewhere [33].

**Carbon source utilization test.** The capacity of the isolate to utilize various carbon compounds, which known *Sphaerotilus* strains commonly utilize, was tested using Armbruster medium [57] as a basal medium. Cultivation was statically done at 25ºC. Utilization was judged by an increase in the optical density (at 660 nm) of the cultures. The capacity to utilize the deicer-related compounds (ethylene glycol, PG, sodium acetate, and sodium formate) was compared with a reference organism, *S. natans* JCM 20382 (= NBRC (IFO) 13543 = ATCC 15291).

## Results

### Biofilm prevalence characteristics

The prevalence of heterotrophic biofilms showed patterns across temporal and spatial gradients (Fig 2). Temporally, sites downstream from the airport had greater biofilm volumes during months with deicer influence (December through May) than those without deicer influence (June through November) (Kruskal-Wallis, $p < 0.05$). Spatially, when compared with DS1, sites farther downstream from the airport generally had lower volumes during months both with and without deicer influence. The upstream site had no notable biofilms regardless of timing.

### Water-quality characteristics

Samples collected concurrently for COD, acetate, and PG showed strong relations between COD and total deicer (i.e., acetate plus PG) concentrations (S1 Fig; $\log_{10}[total\ deicer] = 1.03 * \log_{10}[COD] - 0.47$, $R^2 = 0.93$). Due to this strong relation, as well as the greater expense associated with acetate and PG analyses, COD was used throughout this study as a surrogate for deicer concentrations.

COD concentrations showed notable patterns along both spatial and temporal gradients. Concentrations in grab samples were highest at DS1 and decreased with increasing distance from the airport; lowest concentrations were observed at the upstream site (where values ranged from <8 to 40 mg/L with a median of 16 mg/L; Fig 3A). Highest concentrations at downstream sites were generally observed between mid-December and late March. Similar trends were observed among flow-composite COD samples (S2 Fig). Despite the inherent differences between flow-composite and grab samples, these two types of COD data related well to each other ($\log_{10}[grab\ COD] = 0.97 * \log_{10}[flow\ composite\ COD] + 0.04$, $R^2 = 0.86$; S3 and S4 Figs).

DO concentrations measured during biofilm field surveys also indicated spatial and temporal patterns. Highest overall concentrations were observed upstream from the airport; concentrations at downstream sites typically increased with increasing distance from the airport. Concentrations at US1 peaked in early spring, exhibiting supersaturated concentrations reflective of in-stream photosynthesis [59,60]. During the same time period, DO at all three downstream sites were near their lowest concentrations (Fig 3B).

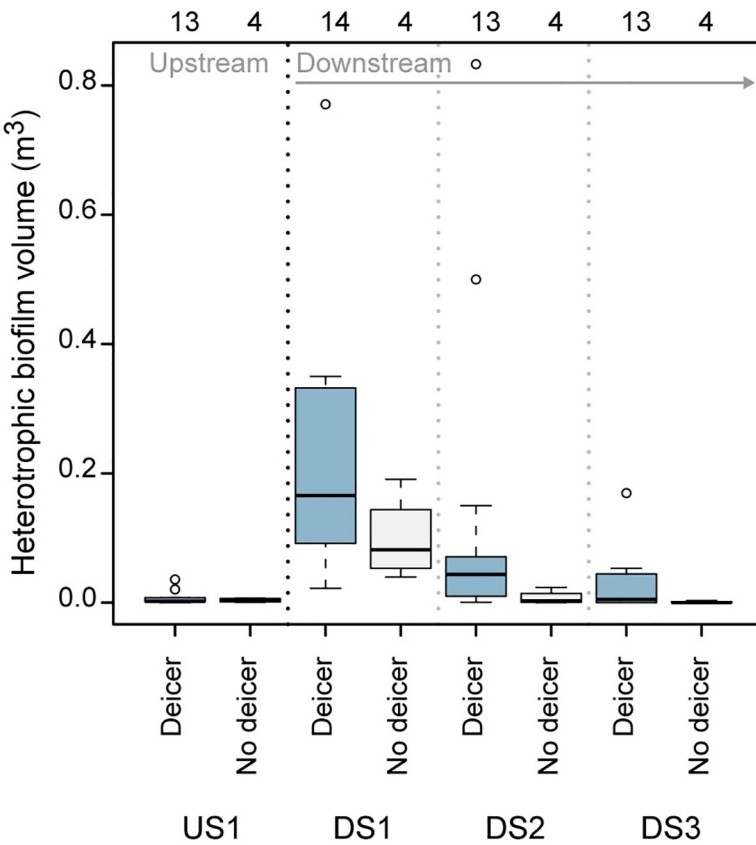

**Fig 2. Volume of biofilms dominated by heterotrophic bacteria during months with deicer influence (December-May) and those without deicer influence (June-November).** Volumes have been normalized to a standard reach size of 50 square meters; number of samples included in each box are given above the graph. Boxplot components represent the following statistics: the midline is the median, the box extends from the 25th to 75th percentiles (i.e., the interquartile range), the whiskers extend to the farthest data point within 1.5 times the interquartile range from the edge of the box in either direction, and circles are individual data points falling beyond that distance.

## Multiple linear regressions

Results of regression analyses for heterotrophic biofilm volumes indicated that variability in volumes was best explained by 2-week, flow-weighted mean COD concentration, together with both site indicator terms ($\log_{10}[biofilm\ volume] = 0.79 * \log_{10}[2\ week,\ flow\ weighted\ mean\ COD] + 1.1 * DS1 + 0.61 * DS2 - 3.9, R^2 = 0.62$; S4 Table). Site indicator terms used here were binary fields denoting whether or not samples were from a given site. Of note, biofilm volumes remained minimal across all sites at mean 2-week, flow-weighted concentrations below 48 mg/ L (Fig 4); biofilm volumes also remained minimal above this value for some samples, the consistency of which increased with increasing distance downstream from the airport.

Results of regression analysis for DO indicated that variability in concentrations was best explained by heterotrophic biofilm volume, 2-week temperature maximum, and 8-week, flow-weighted mean COD concentration ($[dissolved\ oxygen] = -2.2 * \log_{10}[biofilm\ volume] -0.35 * [2\ week\ maximum\ temperature] -4.4 * \log_{10}[8\ week,\ flow\ weighted\ mean\ COD] + 20,\ R^2 = 0.74$; S4 Table).

## Microscopy

Microscopy results provided descriptions of the organisms observed in biofilm samples, with special emphasis paid to the presence/absence of filamentous sheathed bacteria such as

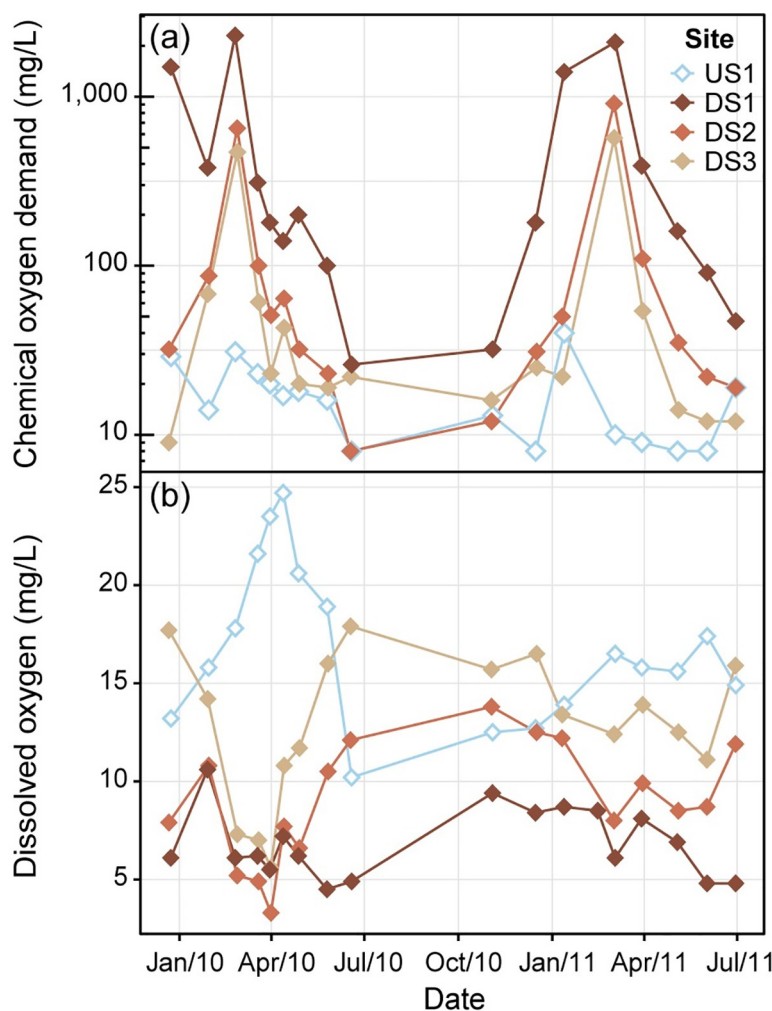

**Fig 3. Water-quality results for grab samples collected during biofilm field investigations.** (a) chemical oxygen demand (COD) concentrations (left-censored values graphed at the reporting limit (of 8 mg/L)) and (b) dissolved oxygen (DO) concentrations.

*Sphaerotilus*, *Leptothrix*, and *Thiothrix*. Results from all samples were binned and graphed (Fig 5). Sheathed bacteria were present in most samples collected from the three downstream sites and were absent in all four of the samples collected at the upstream site. Overall, microscopy results provided a basic context for interpretation of genetic analyses used for this study.

## Microarray

PhyloChip results yielded genetic community structure data with 858 eOTUs for biofilm samples collected across spatial and temporal scales. The resulting genus-level heatmap (Fig 6) shows the abundance of *Sphaerotilus* detected by the PhyloChip was low in many samples, and nonexistent in some, and the abundance of *Thiothrix* ranged from nonexistent to moderate; results were consistent with microscopy assessments (Fig 5), with the exception of the November DS1 sample where sheathed bacteria were observed via microscopy but not via PhyloChip. This difference presumably stems from differences between the 16S sequences present in biofilm sheathed bacteria and those present on the PhyloChip. Although several eOTUs in the

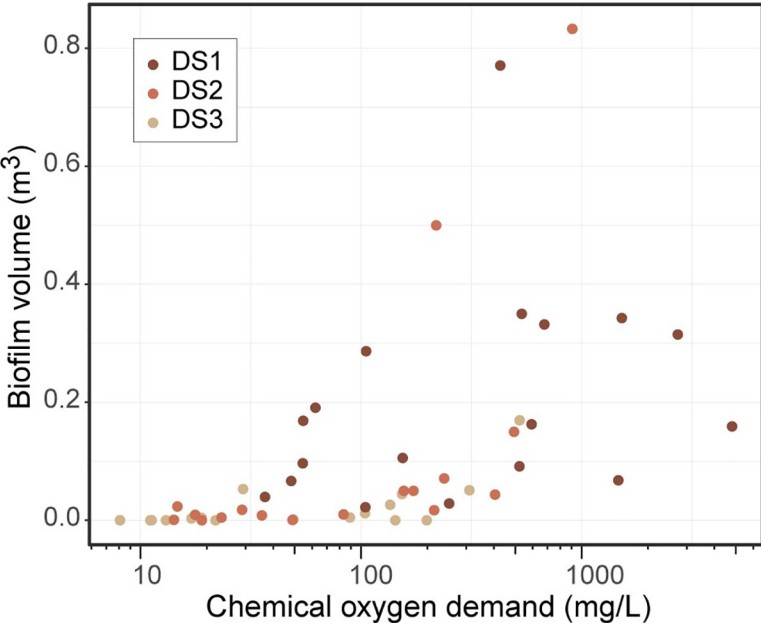

**Fig 4. Comparison of mean 2-week flow-weighted chemical oxygen demand (COD) concentrations and associated biofilm volume measurements.**

family Comamonadaceae were recovered from the PhyloChip analysis results, the genus *Leptothrix*, specifically, was not among them.

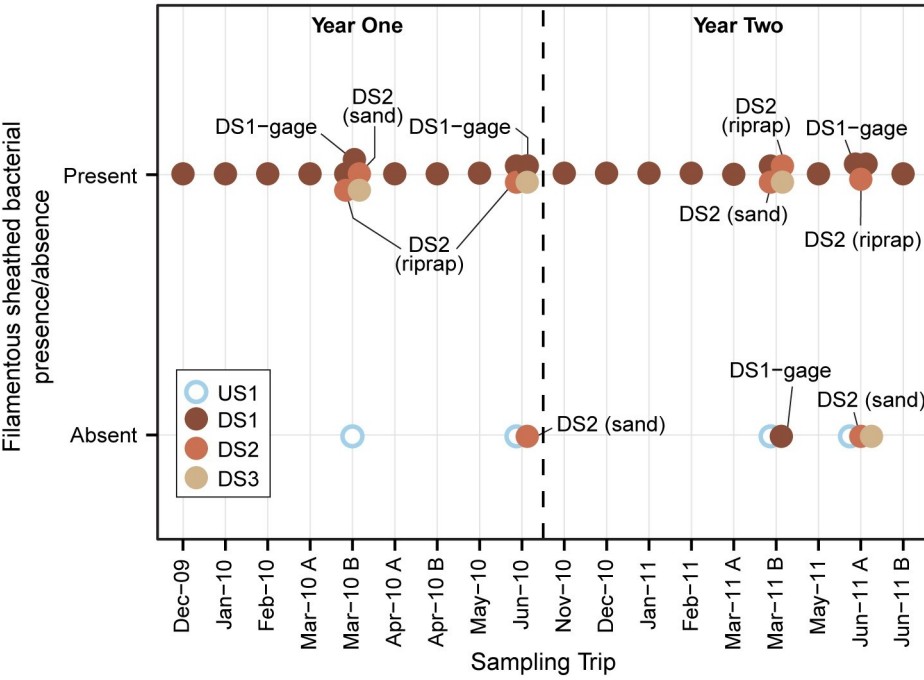

**Fig 5. Microscopy assessments describing the presence or absence of filamentous sheathed bacteria such as *Sphaerotilus*, *Leptothrix*, and *Thiothrix*.** Samples collected from main sites are not labeled; samples from sub-sites are identified with a text label.

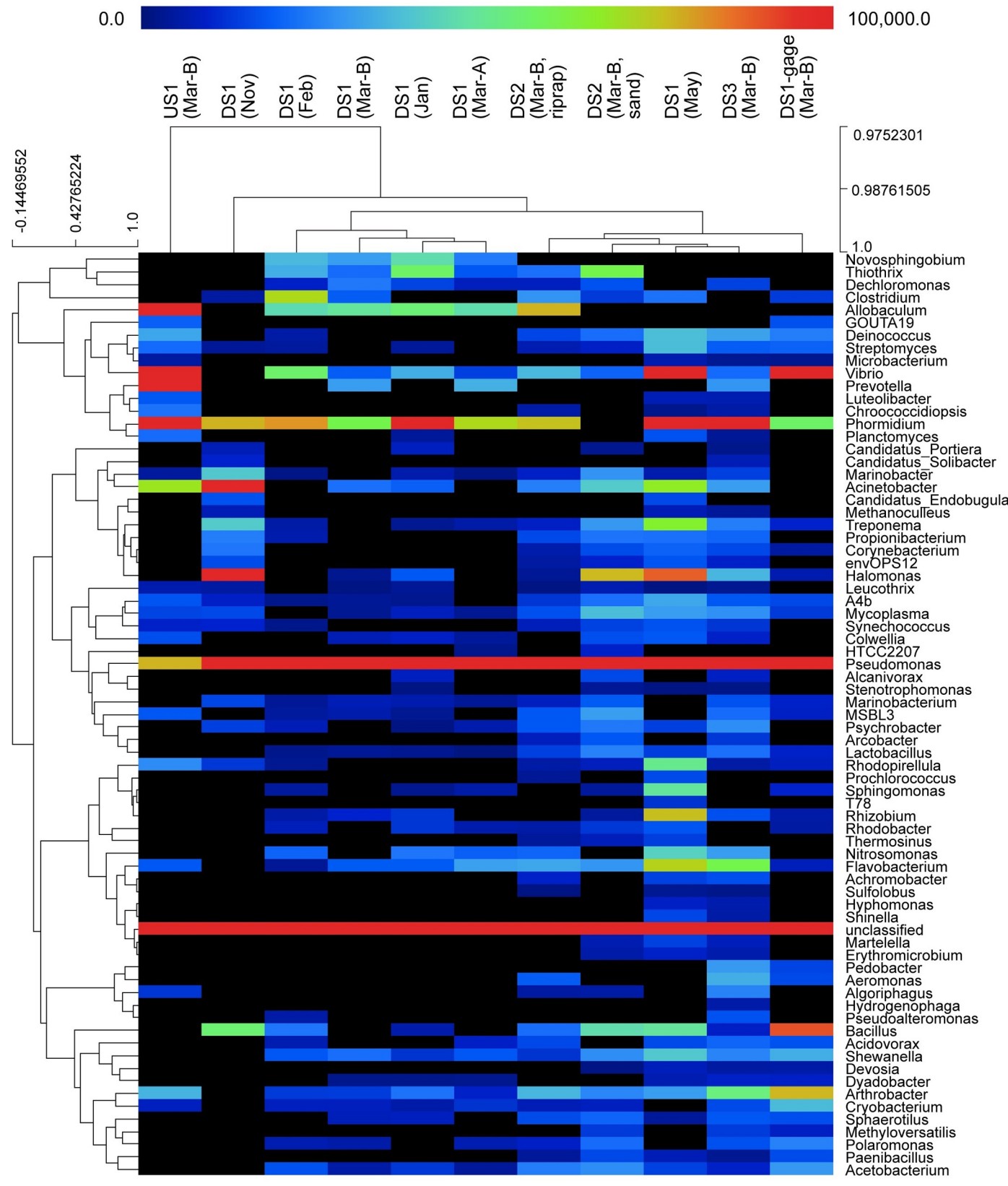

**Fig 6. Heatmap showing genus-level biofilm community composition, based on PhyloChip results.** Dendrograms for the subset of samples analyzed with PhyloChip show relative similarity of biofilm communities at the sample level (columns) and similarity of samples at the genus level (rows). Color scale is based on hybridization scores, which reflect organism abundance; black boxes indicate values of zero. Heatmap created in MeV (MultiExperiment Viewer) in the TM4 software suite [46].

Principal coordinates analysis (PCoA) of PhyloChip data allowed for a more holistic exploration of patterns of ecological change across temporal and spatial scales. PCoA results for samples collected downstream from the airport (during deicer-influenced months) varied primarily along the first axis (Fig 7), with most samples from DS1 clustering towards the left, and all those from DS2 and DS3 clustering towards the right. A notable exception to this trend was the May sample from DS1, which showed greater similarity to March samples collected from DS2 and DS3 than it did to other samples collected at any other time from DS1. Clusters reflecting similarity in microbial community structure corresponded with relatively similar antecedent temperature and COD conditions.

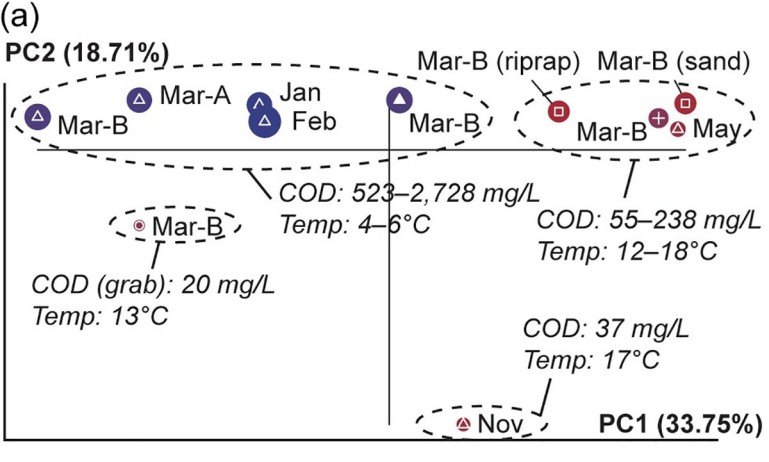

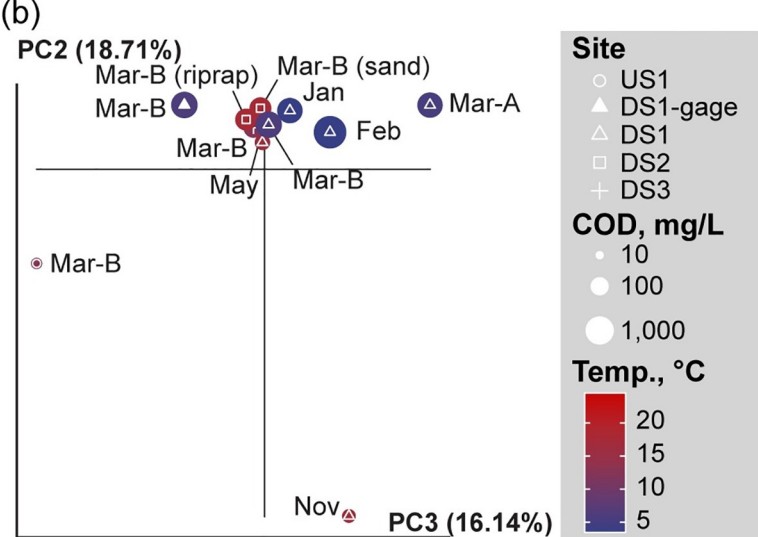

**Fig 7. Principal coordinates analysis plot showing differences in biofilm community composition, based on PhyloChip results.** (a) axes 1 and 2, and (b) axes 2 and 3. White symbol shapes distinguish sampling sites; point size reflects the log-10 mean 2-week flow-weighted COD concentration at each site (except at US1 site, where only grab samples were available); point color reflects the maximum 2-week temperature at each site.

In addition, notable differences were observed between all of these samples and the samples collected upstream from the airport (at US1) and downstream from the airport (at DS1) just prior to the start of deicing.

## Whole metagenome sequencing

Sequence data from the two samples analyzed with WMS were used to generate a consensus sequence for the *sthA* gene present in the samples. The resulting sequence showed 89% concordance with the *Sphaerotilus sthA* sequence described by Suzuki et al. (GenBank AB050640.1; [61]).

Metagenomic whole genome shotgun sequence data were also used to explore community composition through sequence comparison with the NCBI RefSeq Complete Genomes, the One Codex, and targeted loci databases. Metagenomic taxonomy results (RefSeq Complete Genomes and One Codex) indicated microbiome profiles that included *Thiothrix*, *Pseudomonas*, *Janthinobacterium*, and *Hydrogenophaga* (Fig 8). Sequence comparisons with the targeted loci database indicated microbial profiles that included *Spiromyces*, *Flavobacterium*, *Pseudomonas*, *Leptothrix*, and *Polaromonas*. When taken together, the high relative readcount of filamentous sheath bacteria (i.e., *Sphaerotilus*, *Leptothrix*, and *Thiothrix*) were consistent with the presence of sheathed bacteria noted during microscopic analyses of this sample. Results from the genomic databases show substantially higher relative readcounts for *Thiothrix* when compared with *Sphaerotilus* and *Leptothrix*. This difference was likely affected, in part, by sequence representation in the various databases. *Thiothrix* sequences were more populous than those of *Sphaerotilus* and *Leptothrix* in all three databases: RefSeq Complete Genomes (4, 0, and 0 sequences, respectively), One Codex (8, 3, and 2 sequences, respectively), and targeted loci (10, 3, and 3 sequences, respectively).

## Quantitative real-time PCR

Primers designed to target the WMS consensus sequence showed specificity (at Tms of 80˚C and 90˚C) for *Sphaerotilus sthA* sequence and allowed for determinations of *sthA* gene abundance via qPCR.

The ratio of *sthA* DNA to 16S rDNA provided a measure of overall abundance of *Sphaerotilus*, relative to the total bacterial population. This ratio showed strongest positive Spearman rank correlations with heterotrophic biofilm volumes (rho = +0.73), total phosphorus (rho = +0.56), and a number of COD concentration measures—the strongest of which was the 2-week, flow-weighted value (rho = +0.53) (S5 Table). Strongest negative Spearman rank correlations were observed with DO concentration (rho = -0.77), aggregated autotrophic biofilm volume (i.e., the sum of soft algae, transition, and diatom biofilm volumes) (rho = -0.55), and dissolved nitrate+nitrite (rho = -0.51).

## Isolate sequencing analysis, sheath composition analysis, and carbon utilization test

Identical 16S rRNA gene sequences were obtained from all 23 isolates. Sequence analysis and BLAST comparisons of the full, assembled 16S-ITS region of the ribosomal operon to the NCBI nucleotide NR database, indicated that the isolate was a new strain of *Sphaerotilus montanus* (str KMKE). The presence of *sthA* gene was confirmed by amplification and sequencing using developed primers; subsequent comparison showed it to be identical to the WMS consensus sequence from environmental samples. In GC analysis of the hydrolysate of the sheath, glucose and galactosamine were detected showing the biochemical makeup of the isolate sheath to be consistent with that of *S. natans* [54,55]. Supporting the results of the sequencing

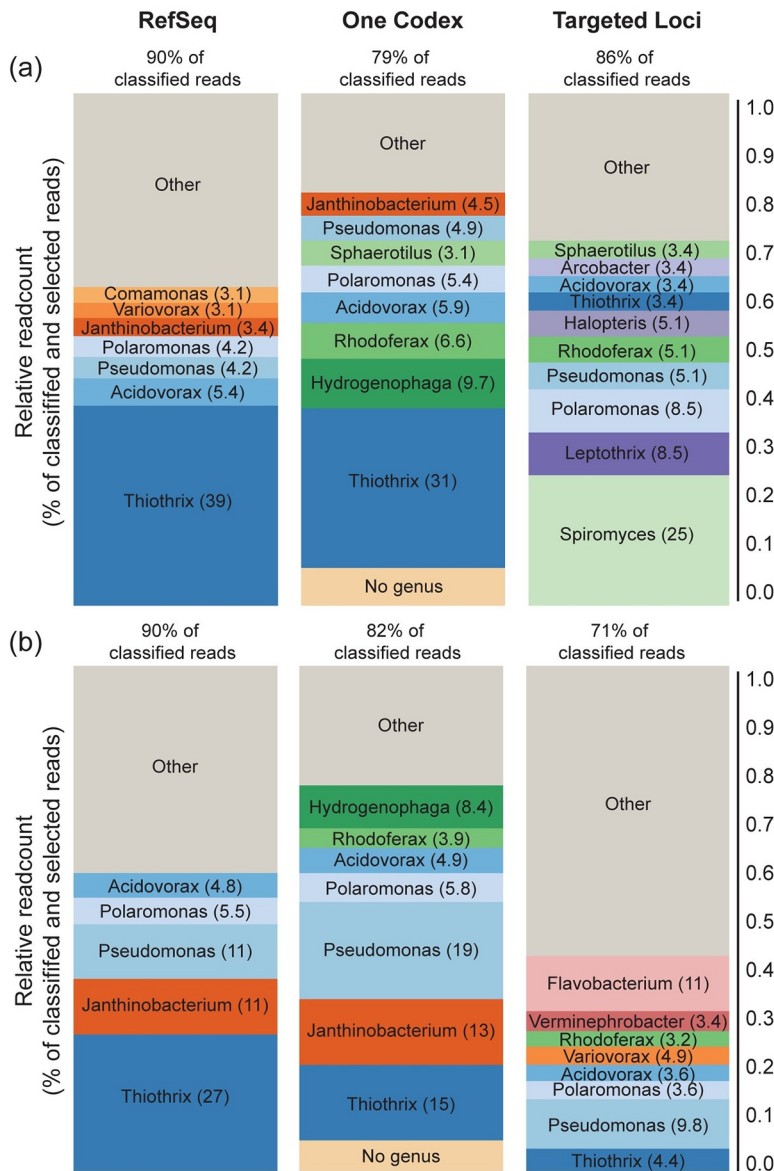

**Fig 8. Normalized genus-level taxonomic alignment of WMS sequence data for samples collected from DS1 on (a) February 24, 2010, and (b) March 18, 2010.** Alignments were limited to those having at least 3% representation with sequences in the RefSeq Complete Genomes, One Codex, and targeted loci databases. Values above bars indicate the percent of reads from the sample that aligned with genus or species level sequences in the indicated database. Values on the bars indicate the relative percentage of aligned sequences that matched to indicated genera.

analysis, the uniqueness of the isolate was revealed by its carbon source utilization pattern (Tables 1 and 2). The carbon sources listed in Table 1 are defined to be commonly utilized by *Sphaerotilus* strains [62]. However, the isolate did not utilize some of these carbon sources indicating the novelty of the isolate. For further characterization, utilization of deicer-related compounds was compared with a well-studied strain of *Sphaerotilus* [54,55]. As shown in Table 2, the isolate utilized PG while the reference strain (*S. natans*) did not. Both organisms utilized acetate.

**Table 1. Typical carbon compound utilization.**

| Carbon source[a] | Isolate | Sphaerotilus strains[b] |
|---|---|---|
| Succinate | + | + |
| Lactate | + | + |
| Pyruvate | + | + |
| Malate | + | + |
| Oxaloacetate | + | + |
| Malonate | - | + |
| Glucose | +[c] | + |
| Sucrose | - | + |
| Maltose | +[c] | + |
| Fructose | - | + |
| Glycerol | + | + |
| Ethanol | + | + |
| Glutamate | + | + |
| Proline | + | + |

[a]Commonly utilized by known *Sphaerotilus* strains.
[b]Data from Gridneva et al., 2011 [62].
[c]Poor growth.

## Discussion

### Biofilm prevalence and response to environmental conditions

Airport biofilms respond to environmental conditions at multiple levels, with differences in biofilm volumes providing the most visible manifestation of this response. In this study, stimulation of biofilm growth by deicers was indicated by the co-occurrence of elevated biofilm abundance and COD concentrations (used here as a surrogate for deicer concentrations) and was supported by regression analysis. This is consistent with an extensive body of literature documenting the frequent occurrence of *Sphaerotilus*-like biofilms in areas of organic pollution [14,16–21,63,64,23]; although explicit investigation of this linkage in the airport setting has been lacking in the literature, it has generally been assumed [63,64].

The observation that minimal biofilm volumes were reliably maintained across downstream sites when antecedent (mean 2-week, flow-weighted) COD concentrations remained below 48 mg/L, and at the upstream site where (grab sample) concentrations remained below 41 mg/L is consistent with recently published observations at a nearby site [64]. This site, Kinnickinnic River at 11[th] Street, is located downstream from the confluence with Wilson Park Creek, and showed no heterotrophic biofilm growth when (grab sample) COD concentrations remained consistently below 66 mg/L.

**Table 2. Deicer-related compound utilization.**

| Compound | Isolate | Sphaerotilus natans[a] |
|---|---|---|
| Diethylene glycol | - | - |
| Propylene glycol | + | - |
| Acetate | + | + |
| Formate | - | - |

[a]JCM 20382 (= NBRC (IFO) 13543 = ATCC 15291).

The differential response of biofilm volumes at different sites to (2-week, flow-weighted) COD concentrations above 48 mg/L is consistent with the importance of site-specific differences observed in the regression. Likewise, although Cincinnati/Northern Kentucky International Airport has shown success in reducing biofilm proliferation by consistently maintaining 5-day biochemical oxygen demand ($BOD_5$) concentrations below 50 mg/L [64], $BOD_5$ alone has not been shown to be a reliable predictor of biofilm growth in all airport settings [64]. For context, according to data in the USGS National Water Information System (NWIS; [65]) from the gaged sites in this study, a $BOD_5$ concentration of 50 mg/L (+/- 5 mg/L) corresponds to COD concentrations ranging from 81 to 140 mg/L. Such site-specific differences likely reflect a combination of distinct physical, chemical, and biological factors affecting the various sites. Recent studies investigating the effects of light, nutrient ratios, and channel depth in this setting found little or no response within the ranges measured [64]. Additional investigations into the effects of environmental variables are warranted, while keeping in mind the important role of organic carbon.

### Biofilm community composition

Traditionally, members of the genus *Sphaerotilus* (*S. natans*, in particular) have been considered the primary organisms driving biofilm formation in areas of organic pollution [14,17–20,23,63,64]. Likewise, microscopic assessments in the current study indicated the relatively consistent presence of sheathed bacteria in heterotrophic biofilms observed at sites receiving deicer runoff; however, these assessments did not include definitive genus-level identifications due to the obvious limitations of visual microscopic identification. Microarray results expanded the descriptive potential of samples by yielding taxonomic assignments together with measures of relative abundances, with microarray results providing insights into community differences across both temporal and spatial scales. WMS results provided additional community composition data for two overlapping samples. Overall, both approaches showed minimal representation by *Sphaerotilus* and indicated that *Thiothrix* may play an important role in the biofilm. However, the relative lack of sequence data available in public databases for organisms in the *Sphaerotilus-Leptothrix* group likely caused some distortion in these results. In the case of microarray, the diversity of 16S sequences for sheathed bacteria were limited by sequence availability at the time of array construction (2006). Likewise, taxonomic hits of community sequences obtained via WMS were skewed by relative population of the databases. Notably, *Thiothrix* appeared most dominant when sequences were compared against genomic databases where there were few or no *Sphaerotilus* or *Leptothrix* sequences. Although microarray and metagenomic sequencing could not definitively identify the predominant sheath bacterium in the biofilm, both approaches showed these biofilms to be diverse communities that included *Thiothrix*, *Pseudomonas*, *Janthinobacterium*, *Hydrogenophaga*, *Spiromyces*, *Flavobacterium*, *Leptothrix*, and *Polaromonas*.

Results from microarray and WMS data yielded some similarities and some differences in the samples analyzed by both. Most notably, microarray showed very high abundance of *Pseudomonas* while WMS indicated abundance to be on par with other represented organisms. Conversely, WMS showed alignment with *Hydrogenophaga*, but this genus was not observed in the microarray dataset. WMS also indicated alignment with *Janthinobacterium*, *Spiromyces*, and *Leptothrix*; however, these organisms were not represented among the eOTUs on the chip. Overall, the predominance of Proteobacteria (and specifically Alphaproteobacteria and Betaproteobacteria) in WMS 16S results were consistent with the broader literature describing similar WMS 16S patterns in typical stream biofilms [25]. Microarray results showing higher than normal representation of Gammaproteobacteria are believed to be an artifact of sequence representation on the chip [25].

In an effort to circumvent issues related to sequence availability in the databases, *sthA* sequences (from the WMS dataset) unique to *Sphaerotilus* were targeted using qPCR. Strong positive correlation between the proportion (relative to total bacteria) of *sthA* DNA and heterotrophic biofilm volume (as well as a complementary negative correlation with autotrophic biofilm volume) indicated the consistent presence and potential importance of these organisms within the heterotrophic biofilm community. Subsequently, a pure strain of filamentous sheathed bacteria from the biofilm was isolated and purified, and additional tests were run to definitively identify the organism; GC analysis of the sheath as well as full-length 16S rDNA sequencing both indicated the closest taxonomic relation to *S. natans* and *S. montanus*, and identified it as a new strain (str KMKE) belonging to *S. montanus* [62]. This identification extends the list of organisms in the *Sphaerotilus-Leptothrix* group known to thrive in high organic settings. When taken together, the similarity of sequences obtained from isolate and WMS datasets as well as the positive correlation of *Sphaerotilus*-specific *sthA* (from qPCR analyses) with biofilm volumes indicated that *S. montanus* was potentially a consistent member of the biofilms observed throughout this study. Based on carbon source utilization, the isolate was distinguishable from other known *Sphaerotilus* strains [62] isolated from ecosystems free from deicer and anti-icer contamination. It is probable that the *Sphaerotilus* strains present in streams contaminated with deicer and anti-icer have distinctive taxonomic features that have adapted to enable them to utilize deicer compounds, such as PG, as carbon sources.

Overall, techniques for accurate and rapid identification of sheathed bacteria are lacking. Traditional microscopic techniques have led to numerous mis-assignments over the years, and genetic taxonomic tools are still being resolved. *Thiothrix* has not traditionally been discussed in these types of settings, and additional study is warranted to determine to what extent these organisms drive biofilm production in areas of organic pollution. Future studies are needed to further characterize new and existing taxonomy and to continue annotating genomic databases.

## Community response to environmental conditions

Despite the limitations of microarray results in identifying specific organisms with rare sequence representation, results did provide detailed community profiles that served as a basis to explore patterns of ecological change across temporal and spatial scales. Clustering patterns in PCoA plots indicated the importance of antecedent COD and temperature characteristics on the community composition of biofilms. Though site-specific differences still appeared to be important at the genetic level, clustering of the DS1 sample from late spring with early spring samples from sites farther downstream, as well as differences observed between the DS1 samples before and during deicer influence further indicated the potential importance of antecedent temperature and COD concentrations on overall biofilm community structure.

In addition to the relations to biofilm volume noted above, qPCR results for *sthA* DNA proportion (relative to bacteria) provided insights into the response of *Sphaerotilus* to changes in environmental conditions. The responses observed were largely in line with expectations. The negative relation of *sthA* with dissolved nitrate+nitrite was consistent with these organisms' known utilization of nitrate as a nitrogen source, and hence higher growth would be expected to translate into greater assimilation [62,66]. Positive relations of *sthA* with COD and total phosphorus were consistent with literature indicating that, in phosphorus-limited, organic-rich systems, increases in phosphorus concentrations correspond with enhanced biofilm growth [64,67]. The strong negative correlation of *sthA* with DO is also consistent with the literature indicating that, despite being an obligate aerobe, *S. natans* preferentially grows at lower DO concentrations [68], thereby conferring a competitive advantage over other aerobic chemoorganoheterotrophs in depressed DO environments [24].

Future studies would benefit from the identification of a gene required for, and exclusively used in filament formation (unlike *sthA*, which is also used in exopolysaccharide generation). Such a gene could be used to identify (via quantitative reverse transcription PCR) the environmental factors that trigger free-swimming cells to switch to a filamentous growth form.

## Dissolved oxygen

DO concentrations are of perennial concern in streams receiving deicer inputs. The observation here that DO concentrations were negatively related to measures of temperature and COD was largely consistent with expectations: with regard to temperature, oxygen solubility is known to increase as temperatures decreases; with regard to COD, depressed DO concentrations in airport receiving streams affected by PDMs and ADAFs are well documented in the literature [10]. Previous research in this same stream system found no correlation between DO and $BOD_5$ concentrations [11]; however, those investigations considered only concurrent measurements. Findings in the current study found antecedent conditions to be more strongly related than concurrent measurements—a finding that indicates the additional influence of a biological mechanism. Influence by a biological mechanism was further supported by the importance of heterotrophic biofilm volume as the third (negative) predictor of DO concentration. Presumably, DO in the stream is consumed by heterotrophic biofilms feeding off the abundant, readily biodegradable organic molecules present, with increased consumption when temperatures are more favorable for them. Typical temperature ranges for *S. montanus* are in the range of 7 to 36 degrees Celsius (optimum: 28–30 degrees Celsius; [62]), however, *Sphaerotilus* species have been shown to adapt to temperatures outside of this range in other environments [20], and have been frequently observed during winter in other areas of organic pollution [17,19]. Enhanced *S. natans* filament formation has been noted in response to low DO concentrations [69], and the literature further indicates that not only does moderate DO depletion stimulate *S. natans* to switch from single-cell to filamentous growth, but that total growth (in both forms) is suppressed at high oxygen concentrations [68]. Previous research has indicated that heterotrophic biofilms likely play a role in DO concentrations in airport receiving streams, but systematic data has previously been lacking to support this.

## Broader implications

Although this study focused on biofilm growth in a stream receiving airport deicing and anti-icing compounds, the implications of this research are far broader. Similar deicers are being suggested for widespread roadway use due performance enhancements and an increased awareness of the persistent ecological damage caused by long-term road salt application [1–5]. Like those found in the airport setting, proposed alternative roadway deicers generally derive their freezing point depression from low-molecular-weight organic molecules [5]. Discussions of the ecological effects stemming from organic roadway deicers have generally been limited to DO depletion, contaminant binding, enhanced algal growth (from phosphorus enrichment), and direct toxicity [70–73]. Heterotrophic biofilm proliferation and the resulting ecological effects have, to date, not been a substantial part of the discussion; findings from this and other studies in the airport literature can help provide decision makers with a fuller picture of the potential effects of widespread alternative deicer use.

## Conclusions

Biofilm volumes in airport receiving streams were minimal below antecedent COD concentrations of 48 mg/L across all sites; above this value site-specific differences became important, with more downstream sites generally having lower biofilm volumes. Biofilms contained a diverse

microbiome, with representation from multiple genera of sheath forming bacteria (*Thiothrix*, *Leptothrix*, and *Sphaerotilus*), among others. This microbiome appeared to shift in composition in relation to antecedent temperature and COD characteristics. Carbon utilization patterns of the isolate *S. montanus* (strain KMKE) showed a unique ability to consume the deicer PG, as compared with closely related organisms, and is believed to have been a consistent and important member of the biofilm community throughout the study, although additional confirmation is warranted. *Sphaerotilus* showed enhanced biofilm representation as DO concentrations decreased. DO concentrations themselves responded to antecedent (but not concurrent) COD concentrations and biofilm volumes, thereby potentially setting up a negative feedback loop.

## Supporting information

**S1 Appendix. Additional details on materials and methods.**
(DOC)

**S1 Table. Site names and characteristics.** Site name, identifiers, drainage area, distance upstream from the Wilson Park Creek at St. Luke's Hospital gage (i.e., DS3-gage) for monitoring sites near Milwaukee Mitchell International Airport (MMIA) in Milwaukee, Wisconsin, USA.
(DOC)

**S2 Table. Physical, water-quality, and biofilm parameters used in stepwise linear regression modeling.**
(DOC)

**S3 Table. Inventory of analyses performed on biofilm samples.** An X indicates that the analytical technique was performed on the sample.
(DOC)

**S4 Table. Predictors selected in multiple linear regressions for explaining variability in heterotroph biofilm prevalence and dissolved oxygen (DO) concentrations.**
(DOC)

**S5 Table. Strongest Spearman rank correlations observed for ratios of *sthA* DNA to 16S rDNA.**
(DOC)

**S1 Fig. Comparison of total deicer (propylene glycol (PG) and acetate) and chemical oxygen demand (COD) concentrations at DS1-gage.** All samples shown here are flow-composite samples. For the purposes of this graph, left-censored deicer concentrations are displayed at one-half the reporting level (5 mg/L for acetate, and 20 mg/L for PG).
(TIF)

**S2 Fig. Chemical oxygen demand (COD) concentrations for flow-composite samples at the three sites downstream from the airport.** Values were measured at DS1 and estimated at DS2 and DS3. Concentrations for grab samples collected upstream from the airport, at US1, are also included for comparison.
(TIF)

**S3 Fig. Chemical oxygen demand (COD) concentrations in flow-composite and grab samples at the three sites downstream from the airport.** Flow-composite sample concentrations were measured at DS1 and estimated at DS2 and DS3.
(TIF)

**S4 Fig. Comparison of chemical oxygen demand (COD) concentrations in grab and (closest) flow-composite samples at the three sites downstream from the airport.** Dashed gray line, 1:1 relation; solid black line, regression (in log10; $R^2$ = 0.86) across all sites and samples. Dark brown points and line, data and regression (in log10; $R^2$ = 0.92) between samples collected at DS1; medium brown points and line, data and regression (in log10; $R^2$ = 0.80) between samples collected at DS2; light brown points and line, data and regression (in log10; $R^2$ = 0.83) between samples collected at DS3.
(TIF)

## Acknowledgments

Special thanks to Greg Failey for programmatic support; to Timothy Hunter for analytical guidance and scoping; to Austin Baldwin, Pete Lenaker, and Troy Rutter for field and data assistance; to Barbara Eikenberry for her guidance on diatoms; and to the Vermont Genetics Network (VGN) Bioinformatics Core for data analysis. The authors would also like to thank the University of Vermont Cancer Center Massively Parallel Sequencing Facility and Vermont Integrative Genomics Resource (VIGR) laboratory for in-kind contributions and devotion to this project. This publication does not necessarily represent the views of NIGMS or NIH but does represent the views of the U.S. Geological Survey. Any use of trade, firm, or product names is for descriptive purposes only and does not imply endorsement by the U.S. Government.

## Author Contributions

**Conceptualization:** Michelle A. Nott, Steven R. Corsi, Scott W. Tighe.

**Formal analysis:** Heather E. Driscoll, Mahesh Vangala.

**Funding acquisition:** Steven R. Corsi.

**Investigation:** Minoru Takeda, Scott W. Tighe.

**Methodology:** Michelle A. Nott, Scott W. Tighe.

**Software:** Mahesh Vangala.

**Writing – original draft:** Michelle A. Nott.

**Writing – review & editing:** Heather E. Driscoll, Scott W. Tighe.

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
