## [Decision Letter · Decision Letter 0]

8 Aug 2019

PONE-D-19-14510

Advanced biofilm analysis in streams receiving organic deicer runoff

PLOS ONE

Dear Ms. Nott,

Thank you for submitting your manuscript to PLOS ONE. After careful consideration, we feel that it has merit but does not fully meet PLOS ONE’s publication criteria as it currently stands. Therefore, we invite you to submit a revised version of the manuscript that addresses the points raised during the review process.

The manuscript describing the study of the effect of airport deicer runoff on stream biofilm by Nott et al. was reviewed by two reviewers. Both agree that the study subject is important and timely. However, both also highlighted a number of important shortcomings in the manuscript, with a number of questions on the methods used and their utility towards meeting the study objectives. There are also a number of problems in the text that take away from the clarity of the study and its potential utility. Both the abstract and the introduction are not well developed or complete. The abstract does not clearly define several key aspects (and their innovative nature) of the study. The final paragraphs of the introduction are a little confusing, the objectives appear to be defined twice and the novelty of the study could be made more evident. The selection of some of the methods and the use of their should also be better defined, and where necessary, properly caveated to identify shortcomings.

I feel that this can be achieved in a revision of these sections and careful editing of the whole manuscript. Both reviewers dedicated significant time to provide constructive comments that should be clearly and completely addressed in the revision.

We would appreciate receiving your revised manuscript by Sep 21 2019 11:59PM. To enhance the reproducibility of your results, we recommend that if applicable you deposit your laboratory protocols in protocols.io, where a protocol can be assigned its own identifier (DOI) such that it can be cited independently in the future. For instructions see: http://journals.plos.org/plosone/s/submission-guidelines#loc-laboratory-protocols

We look forward to receiving your revised manuscript.

Kind regards,

Steven Arthur Loiselle

Academic Editor

PLOS ONE

Journal Requirements:

Reviewers' comments:

Reviewer's Responses to Questions

**Comments to the Author**

1. Is the manuscript technically sound, and do the data support the conclusions?

Reviewer #1: Partly

Reviewer #2: Yes

2. Has the statistical analysis been performed appropriately and rigorously? 

Reviewer #1: N/A

Reviewer #2: Yes

3. Have the authors made all data underlying the findings in their manuscript fully available?

Reviewer #1: Yes

Reviewer #2: Yes

4. Is the manuscript presented in an intelligible fashion and written in standard English?

Reviewer #1: Yes

Reviewer #2: Yes

5. Review Comments to the Author

Reviewer #1: Pg 11, Lines 216-225

I do not think the classification of biofilm into categories is rigorous enough to contribute to the readers’ understanding of the conclusions. Because biofilm can be highly variable and in open systems such as this, heterotrophs will grow concurrently with algae, I don’t see the point in trying to fit the biofilms into a category. Furthermore, I find the calculation of biofilm volume unnecessary and potentially misleading. Why not just list the measured biofilm thickness from each reach as a mean and standard deviation of the measurements from that reach?

Figure 2

Define error bars, box extent, and the dark line (probably the mean…but please say so).

Figure 3(b)

Since the maximum saturation level of dissolved oxygen in water at 0ºC at sea level is only 14.6 mg/l, I think the DO measurement accuracy has some serious problems. Reporting levels of up to 25 mg/l calls the entire data set into question.

Figure 5

Some description of what constitutes heavy, moderate, or rare filamentous bacterial abundance would be useful…photos would be better.

Reviewer #2: General Comments

This manuscript deals with a very interesting topic, the effect of airport deicer runoff on stream biofilm communities. Some interesting results are presented, including the demonstration of a relationship between deicer usage and biofilm volume in a receiving stream, and the isolation of a new strain of Sphaerotilus montanus from deicer-receiving stream biofilms that is capable of using deicer freezing point depressants as carbon sources. The results have broad implications beyond airport deicers. Specifically, the results provide insights into the potential consequences of the use of organic based roadway deicers as alternatives to road salt, which is becoming increasingly common due to the known negative ecological impacts of road salt.

Some issues of concern:

1. My main concern is that several of the methods that were chosen for this study were not well suited to the goals of the study. One goal of the study was to assess variations in the taxonomic composition of stream biofilms. The microarray approach was a poor choice to achieve this goal, as the authors state that “the diversity of 16S sequences were limited by sequence availability at the time of array construction”. The authors would have been much better off using 16S amplicon sequencing to assess the taxonomic composition of stream biofilms. The authors also used metagenomic sequencing, but their application of this approach had several shortcomings. First, they only conducted metagenomic sequencing for two of their samples, and of these two they only presented data for one sample. Therefore, this analysis provides limited insight into the range of samples collected in their study. Secondly, they only used the metagenomic sequencing to assess community composition (see Figure 8) and did not attempt to extract any functional gene data from this data set, which seems like a missed opportunity. If their goal was only to assess community composition, they would have been better off using 16S amplicon sequencing for a larger number of their samples.

2. I have concerns about the accuracy of their qPCR analysis. The authors state that their qPCR assay produced four amplicons with different Tms, with two of the Tms (80C and 90C) being specific for their sthA target. Why would one target produce two amplicons with such different Tms? How can they quantify their targets from this mixture of four amplicons? They need to provide some explanation of this in the manuscript beyond saying that “Careful examination of the qPCR dissociation curves was necessary ….” I read through the Supplementary Information and found their explanation for the qPCR assay to be inadequate. The authors need to provide much more detailed discussion and supporting data to demonstrate the effectiveness and accuracy of their qPCR assay.

3. The abstract does not do an effective job of summarizing the key findings of the study. Specifically:

a. The abstract should clarify that the deicers are composed of “low-molecular-weight organic compounds” as this might not be commonly known.

b. The first sentence of the abstract introduces the topic of deicers, but the effect of deicers on biofilm growth is not mentioned in the abstract. To make this link more clear, perhaps something similar to the following sentence from the Discussion could be added to the abstract: “stimulation of biofilm growth by deicers was suggested by the co-occurrence of elevated biofilm abundance and COD concentrations (used here as a surrogate for deicer concentrations)”.

c. The statement that “site-specific differences became important” is vague and not informative. It seemed like distance downstream from the airport was a key driver. Perhaps this should be mentioned.

d. The abstract ends abruptly. I would suggest some type of concluding sentence.

4. The microarray results do not make much of a contribution to the study. Perhaps they should be removed from the manuscript, or moved to the supplementary material with minimal discussion.

Specific Comments

Line 37 Change “expands” to either “which expands” or “expanding”.

Line 52 What does “This” refer to?

Line 52 Change “impacts” to “impacts of deicers”

Line 52 This sentence states that biofilm proliferation caused by organic deicers is not “generally recognized”. This seems to contradict the first sentence of the abstract, which states that “Prolific heterotrophic biofilm growth is a common occurrence in airport receiving streams containing deicer and anti-icer runoff”.

Line 53 The word “unique” is not appropriate here. Perhaps “useful” would be better.

Lines 246-247 This first sentence is not needed.

Line 253 How were biofilm samples transported and stored?

Lines 260-266 This section of text is not needed here. The section describing each of the methods should clarify how many and which samples were analyzed by that method.

Lines 266-270 These are results and as such should be reported in the Results section.

Line 284-285 This sentence is not needed and should be removed.

Lines 287-289 A mollusk DNA kit seems like an odd choice, especially since there are biofilm specific kits on the market. Why was this kit included in the kits that were tested? Why were biofilm specific kits not considered?

Line 290 Please state explicitly why this kit was chosen. Did it provide the highest yield of the three kits tested?

Line 308 It would be better to refer to his approach here and throughout the manuscript as “metagenomic sequencing” as this is more informative than “massively parallel sequencing”. For example, 16S amplicon sequencing via Illumina could also be referred to as “massively parallel sequencing”.

Line 332 The authors quantified “16S copies” not “16S genomic copies”.

Line 496 Results are presented for only one sample even though two samples were analyzed. Where are the results for the other sample?

Line 562 The statement “Stream biofilm biomass enhancement in response to labile carbon availability is well-documented in the literature” should be supported by some citations.

Line 595 Remove the word “given”.

Figure 4 regression lines should be included in the figure.

6. PLOS authors have the option to publish the peer review history of their article (what does this mean?). If published, this will include your full peer review and any attached files.

Reviewer #1: No

Reviewer #2: No

---

## [Author Response · Author response to Decision Letter 0]

4 Oct 2019

I have attached a document outlining my responses. I have pasted that text here as well (included images did not come through on this paste, please refer to the uploaded document for a full accounting of my response).

PLOS ONE editor email and reviewers' comments:

PONE-D-19-14510

Advanced biofilm analysis in streams receiving organic deicer runoff

PLOS ONE

Dear Ms. Nott,

Thank you for submitting your manuscript to PLOS ONE. After careful consideration, we feel that it has merit but does not fully meet PLOS ONE’s publication criteria as it currently stands. Therefore, we invite you to submit a revised version of the manuscript that addresses the points raised during the review process.

The manuscript describing the study of the effect of airport deicer runoff on stream biofilm by Nott et al. was reviewed by two reviewers. Both agree that the study subject is important and timely. However, both also highlighted a number of important shortcomings in the manuscript, with a number of questions on the methods used and their utility towards meeting the study objectives. There are also a number of problems in the text that take away from the clarity of the study and its potential utility. Both the abstract and the introduction are not well developed or complete. The abstract does not clearly define several key aspects (and their innovative nature) of the study. The final paragraphs of the introduction are a little confusing, the objectives appear to be defined twice and the novelty of the study could be made more evident. The selection of some of the methods and the use of their should also be better defined, and where necessary, properly caveated to identify shortcomings.

Thank you for bringing the issues with the abstract and introduction to our attention. I have revisited both sections and have edited to clarify the wording and messaging within each. 

Further, we understand your concerns regarding the selected methods, particularly in light of Reviewer #2’s criticisms regarding our use of microarray analyses. We have responded in detail to their specific criticisms in our responses below, and we have also added text to the manuscript to explain why microarray was chosen, and what limitations it has.

I feel that this can be achieved in a revision of these sections and careful editing of the whole manuscript. Both reviewers dedicated significant time to provide constructive comments that should be clearly and completely addressed in the revision.

We would appreciate receiving your revised manuscript by Sep 21 2019 11:59PM. To enhance the reproducibility of your results, we recommend that if applicable you deposit your laboratory protocols in protocols.io, where a protocol can be assigned its own identifier (DOI) such that it can be cited independently in the future. For instructions see: http://journals.plos.org/plosone/s/submission-guidelines#loc-laboratory-protocols

 A rebuttal letter that responds to each point raised by the academic editor and reviewer(s). This letter should be uploaded as separate file and labeled 'Response to Reviewers'.

 A marked-up copy of your manuscript that highlights changes made to the original version. This file should be uploaded as separate file and labeled 'Revised Manuscript with Track Changes'.

 An unmarked version of your revised paper without tracked changes. This file should be uploaded as separate file and labeled 'Manuscript'.

We look forward to receiving your revised manuscript.

Kind regards,

Steven Arthur Loiselle

Academic Editor

PLOS ONE

Journal Requirements:

Reviewer's Responses to Questions

Comments to the Author

1. Is the manuscript technically sound, and do the data support the conclusions?

Reviewer #1: Partly

Reviewer #2: Yes

2. Has the statistical analysis been performed appropriately and rigorously? 

Reviewer #1: N/A

Reviewer #2: Yes

3. Have the authors made all data underlying the findings in their manuscript fully available?

Reviewer #1: Yes

Reviewer #2: Yes

4. Is the manuscript presented in an intelligible fashion and written in standard English?

Reviewer #1: Yes

Reviewer #2: Yes

5. Review Comments to the Author

Reviewer #1: Pg 11, Lines 216-225

I do not think the classification of biofilm into categories is rigorous enough to contribute to the readers’ understanding of the conclusions. Because biofilm can be highly variable and in open systems such as this, heterotrophs will grow concurrently with algae, I don’t see the point in trying to fit the biofilms into a category. Furthermore, I find the calculation of biofilm volume unnecessary and potentially misleading. Why not just list the measured biofilm thickness from each reach as a mean and standard deviation of the measurements from that reach?

I agree that biofilms in natural systems are often highly variable, and your confusion at the need and approach to categorizing them is understandable. However, I would contend that categorizing them was necessary for describing the issue at hand, and that our approach was similar in rigor to published protocols for rapid periphyton surveys and accepted for permit compliance with the Michigan Department of Environment, Great Lakes, and Energy (EGLE).

To address the need for categorization, please see the photos below. Photos A-C depict the starkly different biofilm types encountered during our sampling; were we not to categorize the difference between these biofilms, we’d be unable to both describe the proliferation of heterotrophic biofilms and to distinguish them from the diatom and soft algae-dominated biofilms also encountered at the sites. [Photo D represents stream biofilms that are in transition (i.e., where neither heterotrophs nor soft algae are dominant).] To assist in the reader’s understanding of the range of biofilms encountered at these sites, I have included this figure as a new figure (Fig A) in the Appendix. 

Fig A. Photographs of biofilm classes encountered at sites. (a) heterotroph-dominated biofilm (DS1 reach on December 23rd, 2009); (b) diatom-dominated biofilm (covering macrophytes; US1 reach on March 29th, 2011); (c) soft-algae-dominated biofilm (DS3 reach on June 17th, 2010); and (d) transitional biofilm (i.e., mix of soft algae and heterotrophs; DS1 reach on June 18th, 2010).

To address your concerns about the rigor with which these categorizations were made, I’d refer you to the EPA’s 1999 Rapid Bioassessment Protocols [1] which describes “…a field-based rapid survey of periphyton biomass and coarse-level taxonomic composition (e.g., diatoms, filamentous greens, blue-green algae) and requires [sic] little taxonomic expertise.” An update to that protocol was published in 2007 [2], and has been modified (in consultation with Stevenson) for use in heterotrophic biofilm monitoring efforts by Gerald Ford International Airport (GFIA; Grand Rapids, Michigan), and is required for use by their NPDES permit. I have included that protocol as an attachment to this memo. This modified protocol was employed for a manuscript we (USGS and LimnoTech, the contractor that handles the permit compliance monitoring for GFIA) collaborated on a few years ago [3]. The major differences between this protocol and the one employed here:

GFIA protocol This study

Biofilm thicknesses was recorded as binned value Biofilm thickness was recorded as measured value

Biofilms growing on substrates ≤ 2 centimeters were not measured/considered for heterotrophic growth No restrictions were placed on substrate size

Biofilm type categorized in the field Biofilm type categorized in the office, based on characteristics recorded in the field

Biofilm categories included moss, macroalgae, microalgae, and heterotrophic biofilm Biofilm categories included soft algae, diatoms, heterotrophic biofilm, and transition (i.e., mix of soft algae and heterotrophic biofilm)

All biofilm types present were described/measured at all points they occurred The dominant biofilm type at each point was categorized (characteristics for other biofilm types were not considered)

Biofilm density was recorded as a binned value of the number of dots covered in a 50-point gridded bucket An analogous grid-based measure of density was not collected

Biofilm metrics included biofilm extent (i.e., fraction of considered sampling points with biofilm cover of “<0.5 mm” or larger), and biofilm magnitude which is a unitless number reflecting the average bin and occurrence number. Biofilm metrics are reported here as biofilm volume which incorporates measures of extent (i.e., fraction of points of specified biofilm type) and magnitude (i.e., median thickness of specified biofilm type).

It is our belief that none of the differences outlined here significantly jeopardize the rigor of our approach. The manuscript text referencing methods adapted for use in this study have been updated to include the published protocols mentioned here. Furthermore, to more accurately describe the approach given here, I have included wording in the manuscript to describe the categorizations as operational classifications, with heterotrophic biofilms described as those without visual algal representation.

With respect to your comment regarding our use of biofilm volume instead of thickness, I agree that thickness is much easier to conceptualize than volume, however we feel that your suggested use of thickness summary statistics would be less robust than the measures of volume we’re providing. Though harder to conceptualize, the use of biofilm volume allows us to use a single value to encompass both the thickness and extent of the biofilm with a single value—both of which we feel are critical to consider in understanding the issue at hand. Our overall aim is to describe the abundance of heterotrophic biofilm (and compare abundances between sites), and we feel that representing both of those dimensions is important in accurately conveying this information. 

Figure 2

Define error bars, box extent, and the dark line (probably the mean…but please say so).

I have included a text description at the end of the figure caption that explains the various components of the boxplot. 

Figure 3(b)

Since the maximum saturation level of dissolved oxygen in water at 0ºC at sea level is only 14.6 mg/l, I think the DO measurement accuracy has some serious problems. Reporting levels of up to 25 mg/l calls the entire data set into question.

A 14.62 mg/L value assumes that the water body (0ºC, 760 mm barometric pressure) is in perfect equilibrium with the overlying air, but it does not account for the potential contribution of photosynthesis from within the waterbody. In contrast with the overlying air, which is just 21% oxygen, in-stream photosynthesis releases additional 100% oxygen directly into the water column, thereby potentially creating supersaturated conditions. One would expect that such conditions would occur where there is significant photosynthesis and stagnant water (i.e., where water is not quickly releasing the excess oxygen to the overlying air). The highest values observed in this study were at the upstream site (US1), which had just these conditions—significant algal and macrophyte populations and stagnant water. 

At the USGS, we see such values with decent regularity, so it didn’t occur to me that this might be alarming to others. However, these values caused alarm for both this reviewer and for the reviewer selected for the USGS review. In response, I have added some explanatory language and citations into the results section of the report to help put these higher DO values in context. 

Figure 5

Some description of what constitutes heavy, moderate, or rare filamentous bacterial abundance would be useful…photos would be better. Microscopy data were recategorized as presence/absence data to avoid the subjectivity of these qualitative categorizations. Affected figures, text descriptions, and conclusions/comparisons have all been adjusted to account for this change. 

Reviewer #2: General Comments

This manuscript deals with a very interesting topic, the effect of airport deicer runoff on stream biofilm communities. Some interesting results are presented, including the demonstration of a relationship between deicer usage and biofilm volume in a receiving stream, and the isolation of a new strain of Sphaerotilus montanus from deicer-receiving stream biofilms that is capable of using deicer freezing point depressants as carbon sources. The results have broad implications beyond airport deicers. Specifically, the results provide insights into the potential consequences of the use of organic based roadway deicers as alternatives to road salt, which is becoming increasingly common due to the known negative ecological impacts of road salt.

Some issues of concern:

1. My main concern is that several of the methods that were chosen for this study were not well suited to the goals of the study. One goal of the study was to assess variations in the taxonomic composition of stream biofilms. (A)The microarray approach was a poor choice to achieve this goal, as the authors state that “the diversity of 16S sequences were limited by sequence availability at the time of array construction”. The authors would have been much better off using 16S amplicon sequencing to assess the taxonomic composition of stream biofilms. (B)The authors also used metagenomic sequencing, but their application of this approach had several shortcomings. First, they only conducted metagenomic sequencing for two of their samples, and of these two they only presented data for one sample. Therefore, this analysis provides limited insight into the range of samples collected in their study. (C) Secondly, they only used the metagenomic sequencing to assess community composition (see Figure 8) and did not attempt to extract any functional gene data from this data set, which seems like a missed opportunity. If their goal was only to assess community composition, they would have been better off using 16S amplicon sequencing for a larger number of their samples. I have broken this concern down into three separate issues and I’ve inserted letters above to more easily reference them. In order:

 We understand your concerns about the static nature of the PhyloChip sequence composition. However, the second generation PhyloChip contains 297,851 probes complementary to 842 prokaryotic subfamilies and 8,935 operational taxonomic units (OTUs) [4]. It targets multiple hypervariable regions for each OTU, and while sequence databases have grown and evolved during the time since chip construction, we feel that the PhyloChip still adequately provides a broad assessment of basic community composition. Though this is an older technology, PhyloChips continue to be sold (https://www.secondgenome.com/platform/microbiome-technologies/phylochip-community-analysis) and utilized for community assessments [5–8]. Further, the use of 16S amplicon sequencing comes with its own drawbacks where taxonomic profiling is concerned. Notably, 16S amplicon sequencing uses a single or dual hypervariable amplicon, which provides lower-precision data that may be inadequate for positioning within more slowly evolving groups (e.g., gram positive Actinobacteria), thereby making them indistinguishable from one another. Additionally, mismatches between primers and taxa may prevent amplification altogether. We feel that the manuscript would be enhanced if we explicitly acknowledge the drawbacks of microarray and clearly state our reason for choosing it over other available options; these edits have been made. 

 We agree that analyzing more samples using whole metagenomic sequencing would have been preferable. I would agree that this limits the insights that can be drawn from these data; however, the primary purpose of these data was to identify and characterize the sthA sequence for primer development and subsequent analysis via RTqPCR. As a result, instead of running shotgun sequences, we analyzed 200 million reads per sample to achieve metagenomically assembled gene sequences for alignment purposes. Your comment here alerts me to the fact that I didn’t properly convey our objectives for metagenomic sequencing in the manuscript; I have edited the text to correct that. 

 We would agree that functional annotation would have been a valuable addition to this project. We actually did perform some basic functional annotations using Uniprot and NCBI protein databases with both denovo assembly and reference-based approaches, but we did not include them in the manuscript due to the sample number limitations you cited earlier. Given that the we only had whole metagenome sequence data for two samples (one site at two different timepoints), we felt that the functional analysis was best suited to use as a pilot effort for future research. We felt more confident providing the metagenomic community composition data due to the similar/corroborating data available (for that same sample(s)) in the microarray dataset. Please see my response to issue (A) above for our reasoning in not choosing 16S amplicon sequencing.

2. I have concerns about the accuracy of their qPCR analysis. The authors state that their qPCR assay produced four amplicons with different Tms, with two of the Tms (80C and 90C) being specific for their sthA target. Why would one target produce two amplicons with such different Tms? How can they quantify their targets from this mixture of four amplicons? They need to provide some explanation of this in the manuscript beyond saying that “Careful examination of the qPCR dissociation curves was necessary ….” I read through the Supplementary Information and found their explanation for the qPCR assay to be inadequate. The authors need to provide much more detailed discussion and supporting data to demonstrate the effectiveness and accuracy of their qPCR assay. 

We agree that the RTqPCR work needed further explanation; thank you for bringing this to our attention. To answer your first question, amplicons don’t necessarily melt all at once, so peaks may be seen at multiple Tms based on variable sequence contexts within the same gene. Put differently, AT-rich sequences within the gene melt at lower temperatures than GC-rich sections of the same gene; it’s not surprising to us that this phenomenon happened with the target amplicon, given the variability of GC content in the gene. IDT provides a good explanation of why this happens on their website (https://www.idtdna.com/pages/education/decoded/article/interpreting-melt-curves-an-indicator-not-a-diagnosis). You are right, we failed to explain how we dealt with these off-target amplicons in our dataset. Briefly here, to assure that we were amplifying the intended targets, amplicons were run on a gel, and each band was extracted and sequenced. BLAST comparisons were run with NCBI GenBank NT; this identified two amplicons as on-target and two amplicons as off-target. The fraction of the signal resulting from the on-target Tms was determined from the dissociation curve, and that fraction was multiplied by the total genomic number to yield a corrected value. We have updated the text to clarify how it was that we had two Tms for the same target amplicon, as well as to more thoroughly explain how these data were quantified. We are also adjusting the associated data release product to include the data for total genomic number and the fraction of on-target sequences for each sample.

3. The abstract does not do an effective job of summarizing the key findings of the study. Agreed; in addition to addressing the specific issues called out below, I’ve rewritten the abstract to more clearly communicate our major findings.

Specifically:

a. The abstract should clarify that the deicers are composed of “low-molecular-weight organic compounds” as this might not be commonly known. I’ve included text in the abstract to help clarify this point. 

b. The first sentence of the abstract introduces the topic of deicers, but the effect of deicers on biofilm growth is not mentioned in the abstract. To make this link more clear, perhaps something similar to the following sentence from the Discussion could be added to the abstract: “stimulation of biofilm growth by deicers was suggested by the co-occurrence of elevated biofilm abundance and COD concentrations (used here as a surrogate for deicer concentrations)”. Excellent suggestion. I’ve included text in the abstract indicating that COD was used as a surrogate for deicer concentrations, and that biofilm volumes were stimulated by antecedent COD concentrations.

c. The statement that “site-specific differences became important” is vague and not informative. It seemed like distance downstream from the airport was a key driver. Perhaps this should be mentioned. For the purposes of the abstract, I have removed this statement because I am constrained on space and don’t see this as a major finding. However, this same statement is also made within the discussion (i.e., conclusions) section, and, in keeping with the spirit of this comment, I have elaborated on the wording there. 

d. The abstract ends abruptly. I would suggest some type of concluding sentence. I have restructured the abstract to end with a statement about the potential utility of the airport receiving stream as a microcosm for the larger roadway deicing issue. 

4. The microarray results do not make much of a contribution to the study. Perhaps they should be removed from the manuscript, or moved to the supplementary material with minimal discussion. I disagree with this assessment, and have taken it to mean that I need to clarify the text to make the value of these analyses more apparent to the reader. These were the only taxonomic descriptors that were available across both temporal and spatial scales, and the observation of community changes across these scales was a key piece of this study. Edits have been made in areas throughout the manuscript to try to communicate the importance of the results from the microarray analyses.

Specific Comments

Line 37 Change “expands” to either “which expands” or “expanding”. This phrase was removed as part of the abstract rewrite.

Line 52 What does “This” refer to? I see what you mean, that’s very unclear. I’ve clarified the wording to indicate that the consistent, long-term use of organic deicers in the airport setting makes those receiving streams a good ecosystem for characterizing the potential impacts of more widespread (i.e., roadway) use.

Line 52 Change “impacts” to “impacts of deicers” Restructured sentence slightly to say “effects of organic deicers.”

Line 52 This sentence states that biofilm proliferation caused by organic deicers is not “generally recognized”. This seems to contradict the first sentence of the abstract, which states that “Prolific heterotrophic biofilm growth is a common occurrence in airport receiving streams containing deicer and anti-icer runoff”. I was trying to highlight the disconnect between the two communities (roadway deicing and airport deicing communities). The roadway deicing community is aware of the potential DO impacts of spreading organic deicers, but don’t appear to have considered the proliferation of biofilms as a potential impact, even though it’s common in the airport setting. I’ve attempted to clarify my message regarding this disconnect in the text.

Line 53 The word “unique” is not appropriate here. Perhaps “useful” would be better. Nice catch. Switched to “useful.”

Lines 246-247 This first sentence is not needed. I agree; it has been removed. 

Line 253 How were biofilm samples transported and stored? Thank you for catching that omission. I’ve added text in the manuscript to describe this. 

Lines 260-266 This section of text is not needed here. The section describing each of the methods should clarify how many and which samples were analyzed by that method. I agree; it has been removed. Clarifying text and references to Table S3 were added to each of the methods sections, where appropriate. 

Lines 266-270 These are results and as such should be reported in the Results section. I agree, it’s inappropriate to have them here. I don’t see much value in including them in the results either, so I’ve just removed them from the text entirely. 

Line 284-285 This sentence is not needed and should be removed. I agree; it has been removed.

Lines 287-289 A mollusk DNA kit seems like an odd choice, especially since there are biofilm specific kits on the market. Why was this kit included in the kits that were tested? Why were biofilm specific kits not considered? The mollusc kit provided a pre-packaged version of an approach outlined by Zhou [9], which showed high yields in environmental DNA extractions. The described method used use a cetyl trimethyl ammonium bromide (CTAB) detergent followed by a chloroform purification step. Further, the Qiagen DNeasy PowerBiofilm Kit and the Norgen Biofilm DNA Isolation Kit were known to be poorly performing kits that didn’t use CTAB [10]. I’ve changed the wording to emphasize the method used by the kit (in both main text and appendix) and have added reference to the Zhou paper (in the appendix) to emphasize our reasoning for choosing this kit. 

Line 290 Please state explicitly why this kit was chosen. Did it provide the highest yield of the three kits tested? Yes, it did provide the highest yields. I’ve edited the wording in the main text and the appendix to more explicitly state this. 

Line 308 It would be better to refer to his approach here and throughout the manuscript as “metagenomic sequencing” as this is more informative than “massively parallel sequencing”. For example, 16S amplicon sequencing via Illumina could also be referred to as “massively parallel sequencing”. Good point. In response to this issue, I’d like to go a step further, and refer to it a “whole metagenome sequencing” (WMS) to distinguish it from something like 16S amplicon sequencing (i.e., partial metagenome sequencing). As such, “massively parallel sequencing” has been replaced with “whole metagenome sequencing” or “WMS”, as appropriate, throughout the manuscript. 

Line 332 The authors quantified “16S copies” not “16S genomic copies”. The word “genomic” has been removed. 

Line 496 Results are presented for only one sample even though two samples were analyzed. Where are the results for the other sample? Results for just one of the samples were included in the original draft in an effort to rein in the length on an already long manuscript. However, your comment brings our attention to the lack of transparency shown by this decision. As a result, we have updated Fig 8 to include data from both samples; to assure comparability between samples, the sequences for the previously displayed sample were rerun against the current vintage of the databases, and, as such, show differences from our initial results. We have reworked the text to reflect all of the changes discussed here. 

Line 562 The statement “Stream biofilm biomass enhancement in response to labile carbon availability is well-documented in the literature” should be supported by some citations. Agree this sentence deserves citations, however, in re-reading the paragraph I felt this sentence was unnecessary, so I have removed it instead. 

Line 595 Remove the word “given”. It has been removed. 

Figure 4 regression lines should be included in the figure. The developed regression has 3 explanatory variables and therefore cannot be represented on this bivariate scatterplot. I suspect my reference to Figure 4 in the regression sentence (Line 434) likely caused some confusion, so I’ve removed the reference from that sentence and have moved it to (what was previously) the next sentence (Line 435) where the scatterplot is discussed. 

6. PLOS authors have the option to publish the peer review history of their article (what does this mean?). If published, this will include your full peer review and any attached files.

Do you want your identity to be public for this peer review? For information about this choice, including consent withdrawal, please see our Privacy Policy.

Reviewer #1: No

Reviewer #2: No

 

USGS reviewer’s comments (line numbering in Sturman version/PLOS ONE version):

Line 48/Line 46: I find the continued use of the word ‘antecedent’ somewhat confusing here in the abstract. Do you mean ‘before deicer application’? If so, you should state this. That is not what it means. It refers to the time window leading up to biofilm sampling. I have included text to clarify this in the abstract, and I have also revisited the methods and put the word ‘antecedent’ in there to try to clarify and tie the sections/data together within the report. 

Line 66-67/Line 62: Are the deicers applied as neat formulations? If not, what is the typical dilution and subsequent COD? Type I deicer is diluted 50-70%, depending on conditions, but PDM and Type IV anti-icer formulations are applied at full strength. Your comment alerts me to the fact that it would be more relevant to include just the COD concentrations for the applied formulations, so I have edited the text to make this change. 

Line 85/Line 77: “pollution” - An accurate term, but not very descriptive. Maybe something like ‘external organic carbon inputs’ would be more descriptive. Changed “pollution” to “external organic carbon inputs.”

Line 112/Line 102: Change “the stream” to “the receiving stream” This sentence was removed as part of the rewrite of the introduction.

Line 118/Line 108: Change “brought to bear on” to “used to investigate” This sentence was removed as part of the rewrite of the introduction.

Line 118/Line 108: Change “airport setting” to “airport runoff setting” This sentence was removed as part of the rewrite of the introduction.

Line 269/Line 273: “qualitative abundance” - How was this done? A description of how you determined what was ‘heavy’, ‘moderate’, and ‘rare’ would be helpful. [PLOS ONE Reviewer #1 had this same concern (as related to Fig 5), and our response to it is described there.]

Figure 2/Figure 2: You should define the error bars, box extrema, and thick line (mean, median or mode). [PLOS ONE Reviewer #1 had this same concern, and our response to it is described there.]

Figure 3/Figure 3: For part (a), you should try re-plotting on a semi-log plot to see if you get better representation of the lower COD numbers. For part (b), I’m concerned that some of your readings may be erroneously high. The maximum DO at these temperatures should be approximately 13-14 mg/L. for water in contact with atmospheric air. How are you getting readings of >20 mg/L?

Thank you for your suggestion regarding part (a). I have replaced the linear plot with a semi-log plot. I agree the semi-log plot gives much better representation on the lower COD concentration data; this is particularly relevant given the low concentrations at which we saw biofilm response at the downstream sites. 

[Your comments related to part (b) were similar to comments raised by PLOS ONE Reviewer #1; please see my response described there.] 

Line 433-434/Line 434-435: “biofilm volumes remained at or near zero” - I’d be very careful using “at or near zero” here. You should report your detection limit, and then state that the readings were ‘at or near the detection limit’. I understand your concern here. I have switched the wording here from “biofilm volumes remained at or near zero” to “biofilm volumes remained minimal,” which is consistent with the wording used throughout the report when describing these results. However, to address this more specifically I don’t feel it’s appropriate to give a reporting limit for this calculated (volume) value, because the lack of observed heterotrophic biofilms could be due to either biofilms being below the observable limit for thickness (and classification) or because the biofilms were dominated by some other biofilm type(s). In both cases, the fraction multiplier (F_Heterotrophs) in the equation would be 0. To clarify, I have updated the text to explicitly state that in instances where heterotrophic biofilms were not observed in a sample, volumes calculated to zero because of a zero value for the fraction multiplier in the equation.

Your comment did, however, draw my attention to my inappropriate use of zero values for biofilm thicknesses in the dataset; I have adjusted the text in the appendix to describe the detection limit there. I will additionally cascade this change through the associated USGS Data Release report.

References cited:

1. Stevenson RJ, Bahls LL. Periphyton protocols. Revision to Rapid Bioassessment Protocols for Use in Streams and Rivers: Periphyton, Benthic Macroinvertebrates, and Fish. 1999; 

2. Stevenson RJ, Rollins SL. Chapter 34 - Ecological Assessments with Benthic Algae. Methods in Stream Ecology (Second Edition). San Diego: Academic Press; 2007. pp. 785–803. Available: http://www.sciencedirect.com/science/article/pii/B9780123329080500474

3. ACRP. Understanding Microbial Biofilms in Receiving Waters Impacted by Airport Deicing Activities [Internet]. Washington, D.C.: Transportation Research Board; 2014 p. 63. Report No.: 115. Available: http://www.trb.org/ACRP/Blurbs/171576.aspx

4. DeSantis TZ, Brodie EL, Moberg JP, Zubieta IX, Piceno YM, Andersen GL. High-Density Universal 16S rRNA Microarray Analysis Reveals Broader Diversity than Typical Clone Library When Sampling the Environment. Microb Ecol. 2007;53: 371–383. doi:10.1007/s00248-006-9134-9

5. Berendsen RL, Vismans G, Yu K, Song Y, Jonge R de, Burgman WP, et al. Disease-induced assemblage of a plant-beneficial bacterial consortium. ISME J. 2018;12: 1496–1507. doi:10.1038/s41396-018-0093-1

6. Mapelli F, Marasco R, Fusi M, Scaglia B, Tsiamis G, Rolli E, et al. The stage of soil development modulates rhizosphere effect along a High Arctic desert chronosequence. ISME J. 2018;12: 1188–1198. doi:10.1038/s41396-017-0026-4

7. Yang C, Powell CA, Duan Y, Shatters R, Fang J, Zhang M. Deciphering the Bacterial Microbiome in Huanglongbing-Affected Citrus Treated with Thermotherapy and Sulfonamide Antibiotics. PLOS ONE. 2016;11: e0155472. doi:10.1371/journal.pone.0155472

8. Piceno YM, Pecora-Black G, Kramer S, Roy M, Reid FC, Dubinsky EA, et al. Bacterial community structure transformed after thermophilically composting human waste in Haiti. PLOS ONE. 2017;12: e0177626. doi:10.1371/journal.pone.0177626

9. Zhou J, Bruns MA, Tiedje JM. DNA recovery from soils of diverse composition. Appl Environ Microbiol. 1996;62: 316–322. 

10. Lear G, Dong Y, Lewis G. Comparison of methods for the extraction of DNA from stream epilithic biofilms. Antonie van Leeuwenhoek. 2010;98: 567–571. doi:10.1007/s10482-010-9464-y

---

## [Decision Letter · Decision Letter 1]

27 Nov 2019

PONE-D-19-14510R1

Advanced biofilm analysis in streams receiving organic deicer runoff

PLOS ONE

Dear Ms. Nott,

Thank you for submitting your manuscript to PLOS ONE. After careful consideration, we feel that it has merit but does not fully meet PLOS ONE’s publication criteria as it currently stands. Therefore, we invite you to submit a revised version of the manuscript that addresses the minor points raised during the review process. These are limited to those suggested by Reviewer 2 regarding the primer design and a couple additional comments.

We would appreciate receiving your revised manuscript by Jan 11 2020 11:59PM. To enhance the reproducibility of your results, we recommend that if applicable you deposit your laboratory protocols in protocols.io, where a protocol can be assigned its own identifier (DOI) such that it can be cited independently in the future. For instructions see: http://journals.plos.org/plosone/s/submission-guidelines#loc-laboratory-protocols

We look forward to receiving your revised manuscript.

Kind regards,

Steven Arthur Loiselle

Academic Editor

PLOS ONE

Reviewers' comments:

Reviewer's Responses to Questions

**Comments to the Author**

1. If the authors have adequately addressed your comments raised in a previous round of review and you feel that this manuscript is now acceptable for publication, you may indicate that here to bypass the “Comments to the Author” section, enter your conflict of interest statement in the “Confidential to Editor” section, and submit your "Accept" recommendation.

Reviewer #1: All comments have been addressed

Reviewer #2: (No Response)

2. Is the manuscript technically sound, and do the data support the conclusions?

Reviewer #1: Yes

Reviewer #2: Yes

3. Has the statistical analysis been performed appropriately and rigorously? 

Reviewer #1: Yes

Reviewer #2: Yes

4. Have the authors made all data underlying the findings in their manuscript fully available?

Reviewer #1: Yes

Reviewer #2: Yes

5. Is the manuscript presented in an intelligible fashion and written in standard English?

Reviewer #1: Yes

Reviewer #2: Yes

6. Review Comments to the Author

Reviewer #1: (No Response)

Reviewer #2: I served as reviewer 2 on the prior submission of this manuscript. In my opinion the authors have effectively addressed the issues raised by myself and reviewer 1. The current version of the manuscript is significantly improved over the prior version. In particular, the authors' explanations of the use of the Phylo Chip and the metagenomic sequencing data are very helpful.

My only specific criticism is that the section on primer design (lines 323 to 28) does not provide enough detail, e.g. software used, the specific primer sequences, the size of the amplicon, etc.

I have a few other minor comments that the authors could address:

Line 32 Change "with one previously identified" to "with a previously identified sthA sequence"

Line 33 Replace "RTqPCR" with "quantitative PCR".

Line 35 The phrase "stimulated by antecedent chemical oxygen demand concentrations" does not specify a positive or negative relationship. I would change this phrase either to "stimulated by elevated antecedent chemical oxygen demand concentrations" or "was positively correlated with antecedent chemical oxygen demand concentrations".

Line 39 Remove the word "characteristics".

Line 45 Remove the word "characteristics".

Line 106 Remove the word "on".

Line 262 Change "was" to "were".

Line 265 The phrase "described lab temperatures" is unclear. Perhaps "temperatures described above" would be more clear.

Line 337 Replace "RTqPCR" with "Quantitative PCR".

Line 338 Change "estimate" to "quantify".

7. PLOS authors have the option to publish the peer review history of their article (what does this mean?). If published, this will include your full peer review and any attached files.

Reviewer #1: No

Reviewer #2: No

---

## [Author Response · Author response to Decision Letter 1]

18 Dec 2019

6. Review Comments to the Author

Reviewer #1: (No Response)

Reviewer #2: I served as reviewer 2 on the prior submission of this manuscript. In my opinion the authors have effectively addressed the issues raised by myself and reviewer 1. The current version of the manuscript is significantly improved over the prior version. In particular, the authors' explanations of the use of the Phylo Chip and the metagenomic sequencing data are very helpful.

My only specific criticism is that the section on primer design (lines 323 to 28) does not provide enough detail, e.g. software used, the specific primer sequences, the size of the amplicon, etc. Thank you for bringing this to our attention. Additional detail describing primer design has been included in both the main text and appendix. 

I have a few other minor comments that the authors could address:

Line 32 Change "with one previously identified" to "with a previously identified sthA sequence" I agree, this is much clearer; this edit has been made.

Line 33 Replace "RTqPCR" with "quantitative PCR". The text has been updated to read ‘quantitative real-time PCR.’ The addition of ‘real-time’ to this suggestion was an important distinction for one of the coauthors and doesn’t change the meaning here. 

Overall, I understand that this was likely just a request to avoid using potentially confusing acronyms in the abstract, however, it made me wonder about our use of RTqPCR as an acronym for quantitative real-time PCR. The change from qPCR to RTqPCR was made during our last round of edits at the behest of a coauthor (i.e., not one of the reviewers). However, in looking at the literature it appears that we were incorrect in making this change, and that the scientific agreement for this acronym (i.e., RTqPCR) is to use it for referring to reverse transcription qPCR [1]. As such, all instances of RTqPCR in this manuscript (and the supporting Data Release) have been changed to qPCR.

Line 35 The phrase "stimulated by antecedent chemical oxygen demand concentrations" does not specify a positive or negative relationship. I would change this phrase either to "stimulated by elevated antecedent chemical oxygen demand concentrations" or "was positively correlated with antecedent chemical oxygen demand concentrations". Good catch; thank you for the suggested wording. I’ve edited text to match your first suggestion here.

Line 39 Remove the word "characteristics". This edit has been made.

Line 45 Remove the word "characteristics". This edit has been made.

Line 106 Remove the word "on". Good catch; this edit has been made.

Line 262 Change "was" to "were". Good catch; this edit has been made.

Line 265 The phrase "described lab temperatures" is unclear. Perhaps "temperatures described above" would be more clear. I agree, this is much clearer; this edit has been made.

Line 337 Replace "RTqPCR" with "Quantitative PCR". The text has been updated to read ‘Quantitative real-time PCR.’ I have also updated the RTqPCR header in the Results section to read the same (line 541). 

Line 338 Change "estimate" to "quantify". This edit has been made.

---

## [Editor Report · Decision Letter 2]

23 Dec 2019

Advanced biofilm analysis in streams receiving organic deicer runoff

PONE-D-19-14510R2

Dear Dr. Nott,

We are pleased to inform you that your manuscript has been judged scientifically suitable for publication and will be formally accepted for publication once it complies with all outstanding technical requirements.

With kind regards,

Steven Arthur Loiselle

Academic Editor

PLOS ONE
---

## [Editor Report · Acceptance letter]

30 Dec 2019

PONE-D-19-14510R2 

Advanced biofilm analysis in streams receiving organic deicer runoff 

Dear Dr. Nott:

I am pleased to inform you that your manuscript has been deemed suitable for publication in PLOS ONE. Congratulations! Your manuscript is now with our production department. 

With kind regards,

on behalf of

Dr. Steven Arthur Loiselle 

Academic Editor

PLOS ONE